# EUREKA: HUMAN-LEVEL REWARD DESIGN VIA CODING LARGE LANGUAGE MODELS

## ABSTRACT

Large Language Models (LLMs) have excelled as high-level semantic planners for sequential decision-making tasks. However, harnessing them to learn complex low-level manipulation tasks, such as dexterous pen spinning, remains an open problem. We bridge this fundamental gap and present EUREKA, a human-level reward design algorithm powered by LLMs. EUREKA exploits the remarkable zero-shot generation, code-writing, and in-context improvement capabilities of state-of-the-art LLMs, such as GPT-4, to perform evolutionary optimization over reward code. The resulting rewards can then be used to acquire complex skills via reinforcement learning. Without any task-specific prompting or pre-defined reward templates, EUREKA generates reward functions that outperform expert human-engineered rewards. In a diverse suite of 29 open-source RL environments that include 10 distinct robot morphologies, EUREKA outperforms human experts on **83%** of the tasks, leading to an average normalized improvement of **52%**. The generality of EUREKA also enables a new gradient-free in-context learning approach to reinforcement learning from human feedback (RLHF), readily incorporating human inputs to improve the quality and the safety of the generated rewards without model updating. Finally, using EUREKA rewards in a curriculum learning setting, we demonstrate for the first time, a simulated Shadow Hand capable of performing pen spinning tricks, adeptly manipulating a pen in circles at rapid speed.

## 1 INTRODUCTION

Large Language Models (LLMs) have excelled as high-level semantic planners for robotics tasks (Ahn et al., 2022; Singh et al., 2023), but whether they can be used to learn complex low-level manipulation tasks, such as dexterous pen spinning, remains an open problem. Existing attempts require substantial domain expertise to construct task prompts or learn only simple skills, leaving a substantial gap in achieving human-level dexterity (Yu et al., 2023; Brohan et al., 2023).

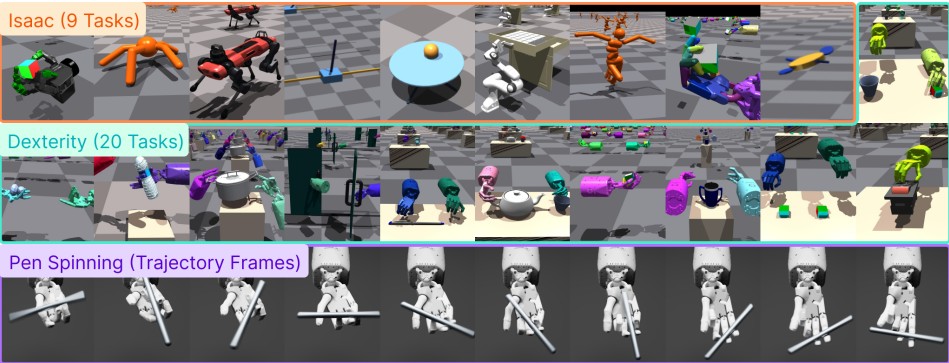

Figure 1: EUREKA generates human-level reward functions across diverse robots and tasks. Combined with curriculum learning, EUREKA for the first time, unlocks rapid pen-spinning capabilities on an anthropomorphic five-finger hand.

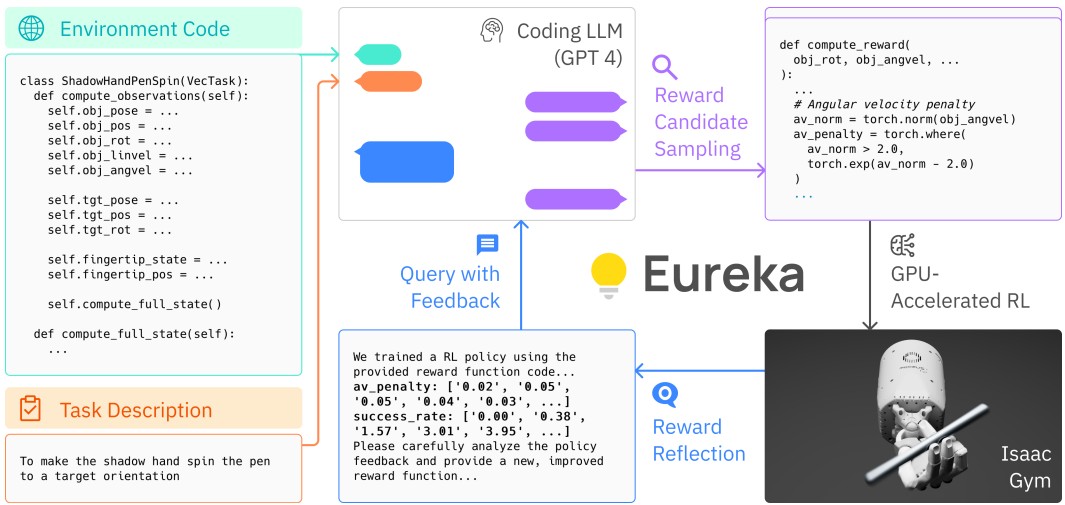

Figure 2: EUREKA takes unmodified environment source code and language task description as context to zero-shot generate executable reward functions from a coding LLM. Then, it iterates between reward sampling, GPU-accelerated reward evaluation, and reward reflection to progressively improve its reward outputs.

On the other hand, reinforcement learning (RL) has achieved impressive results in dexterity (Andrychowicz et al., 2020; Handa et al., 2023) as well as many other domains-if the human designers can carefully construct reward functions that accurately codify and provide learning signals for the desired behavior; likewise, many real-world RL tasks admit sparse rewards that are difficult for learning, necessitating reward shaping that provides incremental learning signals. Despite their fundamental importance, reward functions are known to be notoriously difficult to design in practice (Russell & Norvig, 1995; Sutton & Barto, 2018); a recent survey conducted finds 92% of polled reinforcement learning researchers and practitioners report manual trial-and-error reward design and 89% indicate that their designed rewards are sub-optimal (Booth et al., 2023) and lead to unintended behavior (Hadfield-Menell et al., 2017).

Given the paramount importance of reward design, we ask whether it is possible to develop a *universal* reward programming algorithm using state-of-the-art coding LLMs, such as GPT-4. Their remarkable abilities in code writing, zero-shot generation, and in-context learning have previously enabled effective programmatic agents (Shinn et al., 2023; Wang et al., 2023a). Ideally, this reward design algorithm should achieve human-level reward generation capabilities that scale to a broad spectrum of tasks, including dexterity, automate the tedious trial-and-error procedure without human supervision, and yet be compatible with human oversight to assure safety and alignment.

We introduce **E**volution-driven **U**niversal **RE**ward **K**it for **A**gent (EUREKA), a novel reward design algorithm powered by coding LLMs with the following contributions:

1. **Achieves human-level performance on reward design** across a diverse suite of 29 open-sourced RL environments that include 10 distinct robot morphologies, including quadruped, quadcopter, biped, manipulator, as well as several dexterous hands; see Fig. 1. Without any task-specific prompting or reward templates, EUREKA autonomously generates rewards that outperform expert human rewards on **83%** of the tasks and realizes an average normalized improvement of **52%**.

2. **Solves dexterous manipulation tasks that were previously not feasible by manual reward engineering**. We consider pen spinning, in which a five-finger hand needs to rapidly rotate a pen in pre-defined spinning configurations for as many cycles as possible. Combining EUREKA with curriculum learning, we demonstrate for the first time rapid pen spinning maneuvers on a simulated anthropomorphic Shadow Hand (see Figure 1 bottom).

3. **Enables a new *gradient-free* in-context learning approach to reinforcement learning from human feedback (RLHF)** that can generate more performant and human-aligned reward functions based on various forms of human inputs without model updating. We demonstrate that EUREKA can readily benefit from and improve upon existing human reward functions. Likewise, we

showcase EUREKA's capability in using purely textual feedback to generate progressively more human-aligned reward functions.

Unlike prior work L2R on using LLMs to aid reward design (Yu et al., 2023), EUREKA is completely free of task-specific prompts, reward templates, as well as few-shot examples. In our experiments, EUREKA significantly outperforms L2R due to its ability to generate free-form, expressive reward programs. EUREKA's generality is made possible through three key algorithmic design choices: environment as context, evolutionary search, and reward reflection. First, by taking the **environment source code as context**, EUREKA can zero-shot generate executable reward functions from the backbone coding LLM (GPT-4). Then, EUREKA substantially improves the quality of its rewards by performing **evolutionary search**, iteratively proposing batches of reward candidates and refining the most promising ones within the LLM context window. This in-context improvement is made effective via **reward reflection**, a textual summary of the reward quality based on policy training statistics that enables automated and targeted reward editing; see Fig. 13 for an example of EUREKA zero-shot reward as well as various improvements accumulated during its optimization. To ensure that EUREKA can scale up its reward search to maximum potential, EUREKA evaluates intermediate rewards using GPU-accelerated distributed reinforcement learning on IsaacGym (Makoviychuk et al., 2021), which offers up to three orders of magnitude in policy learning speed, making EUREKA an extensive algorithm that scales naturally with more compute. See Fig. 2 for an overview. We are committed to open-sourcing all prompts, environments, and generated reward functions to promote further research on LLM-based reward design.

## 2 PROBLEM SETTING AND DEFINITIONS

The goal of reward design is to return a shaped reward function for a ground-truth reward function that may be difficult to optimize directly (e.g., sparse rewards); this ground-truth reward function may only be accessed via queries by the designer. We first introduce the formal definition from Singh et al. (2010), which we then adapt to the program synthesis setting, which we call *reward generation*.

**Definition 2.1.** (Reward Design Problem (Singh et al., 2010)) A *reward design problem* (RDP) is a tuple $P = \langle M, \mathcal{R}, \pi_M, F \rangle$, where $M = (S, A, T)$ is the *world model* with state space $S$, action space $A$, and transition function $T$. $\mathcal{R}$ is the space of reward functions; $\mathcal{A}_M(\cdot) : \mathcal{R} \to \Pi$ is a learning algorithm that outputs a policy $\pi : S \to \Delta(A)$ that optimizes reward $R \in \mathcal{R}$ in the resulting *Markov Decision Process* (MDP), $(M, R)$; $F : \Pi \to \mathbb{R}$ is the *fitness* function that produces a scalar evaluation of any policy, which may only be accessed via policy queries (i.e., evaluate the policy using the ground truth reward function). In an RDP, the goal is to output a reward function $R \in \mathcal{R}$ such that the policy $\pi := \mathcal{A}_M(R)$ that optimizes $R$ achieves the highest fitness score $F(\pi)$.

**Reward Generation Problem.** In our problem setting, every component within a RDP is specified via code. Then, given a string $l$ that specifies the task, the objective of the reward generation problem is to output a reward function code $R$ such that $F(\mathcal{A}_M(R))$ is maximized.

## 3 METHOD

EUREKA consists of three algorithmic components: 1) environment as context that enables zero-shot generation of executable rewards, 2) evolutionary search that iteratively proposes and refines reward candidates, and 3) reward reflection that enables fine-grained reward improvement. See Alg. 1 for pseudocode and App. G for EUREKA example outputs; all prompts are included in App. B.

### 3.1 ENVIRONMENT AS CONTEXT

Reward design requires the environment specification to be provided to the LLM. We propose directly feeding the raw environment code (without the reward code, if exists) as context. That is, the LLM will quite literally take $M$ as context. This is intuitive for two reasons: First, coding LLMs are trained on native code written in existing programming languages, so we should expect their code generation capability to be stronger when we directly allow them to compose in the style and syntax they are trained on. Second, and more fundamentally, the environment source code typically **reveals what the environment semantically entails and which variables can and should be used** to compose

a reward function for the specified task. Leveraging these insights, EUREKA instructs the coding LLM to directly return executable Python code with only generic reward design and formatting tips, such as exposing individual components in the reward as a dictionary output (for reasons that will be apparent in Sec. 3.3). This procedure is maximally scalable as the environment source code, by construction, must exist. see App. E for details.

Remarkably, with only these minimal instructions, EUREKA can already zero-shot generate plausibly-looking rewards in diverse environments in its first attempts. An example EUREKA output is shown in Fig. 13. As seen, EUREKA adeptly composes over existing observation variables (e.g., `fingertip_pos`) in the provided environment code and produces a competent reward code – all without any environment-specific prompt engineering or reward templating. On the first try, however, the generated reward may not always be executable, and even if it is, it can be quite sub-optimal with respect to the task fitness metric $F$. While we can

---

**Algorithm 1** EUREKA

1: **Require**: Task description $l$, environment code $M$,
   coding LLM `LLM`, fitness function $F$, initial prompt `prompt`
2: **Hyperparameters**: search iteration $N$, iteration batch size $K$
3: **for** N iterations **do**
4:      `// Sample K reward code from LLM`
5:      $R_1, ..., R_k \sim$ `LLM`$(l, M,$ `prompt`$)$
6:      `// Evaluate reward candidates`
7:      $s_1 = F(R_1), ..., s_K = F(R_K)$
8:      `// Reward reflection`
9:      `prompt := prompt :` $\texttt{Reflection}(R_{best}^n, s_{best}^n)$,
       where $best = \arg\max_k s_1, ..., s_K$
10: `// Update Eureka reward`
11:     $R_{\text{Eureka}}, s_{\text{Eureka}} = (R_{best}^n, s_{best}^n)$,    if $s_{best}^n > s_{\text{Eureka}}$
12: **Output**: $R_{\text{Eureka}}$

---

improve the prompt with task-specific formatting and reward design hints, doing so does not scale to new tasks and hinders the overall generality of our system. How can we effectively overcome the sub-optimality of single-sample reward generation?

## 3.2 EVOLUTIONARY SEARCH

In this section, we will demonstrate how evolutionary search presents a natural solution that addresses the aforementioned execution error and sub-optimality challenges. In each iteration, EUREKA samples several independent outputs from the LLM (Line 5 in Alg. 1). Since the generations are i.i.d, the probability that *all* reward functions from an iteration are buggy *exponentially* decreases as the number of samples increases. We find that for all environments we consider, even sampling just a handful (16) of outputs contains at least one executable reward code in the first iteration.

Giving executable reward functions from an earlier iteration, EUREKA performs in-context *reward mutation*, proposing a new improved reward function from an existing one based on textual feedback. Given the instruction-following and in-context improvement capabilities of LLMs, EUREKA achieves this by simply specifying the mutation operator as a text prompt that suggests a few general ways to modify an existing reward code based on a textual summary of policy training (Sec. 3.3). Several illustrative reward modifications are visualized in Fig. 13. Through mutation, a new EUREKA iteration will take the best-performing reward from the previous iteration as context and generate $K$ more i.i.d reward outputs from the LLM. This iterative optimization continues until a specified number of iterations has been reached. Finally, we perform multiple random restarts to find better maxima; this is a standard strategy in global optimization. In all our experiments, EUREKA conducts 5 independent runs per environment, and for each run, searches for 5 iterations with $K = 16$ samples per iteration.

## 3.3 REWARD REFLECTION

In order to ground the in-context reward mutation, we must be able to put into words the quality of the generated rewards. As we can query the task fitness function $F$ on the resulting policies, a simple strategy is to just provide this numerical score as the reward evaluation. While serving as the holistic ground-truth metric, the task fitness function itself lacks in credit assignment, providing no useful information on *why* a reward function works or not. To provide a more intricate and targeted diagnosis for the rewards, we propose to construct automated feedback that summarizes the policy training dynamics in texts. Specifically, given that EUREKA reward functions are asked to expose their individual components in the reward program (e.g., `reward_components` in Fig. 13),

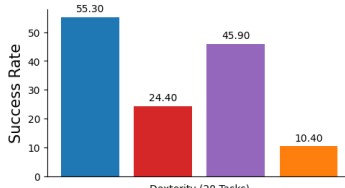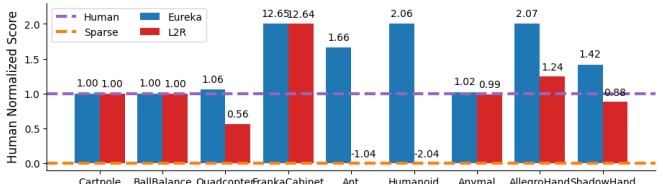

Figure 3: EUREKA outperforms Human and L2R across all tasks. In particular, EUREKA realizes much greater gains on high-dimensional dexterity environments.

we track the scalar values of all reward components at intermediate policy checkpoints throughout training. For instance, consider the illustrative example in Fig. 2, where the snapshot values of `av_penalty` are provided as a list in the reward feedback.

This reward reflection procedure, though simple to construct, is important due to the algorithm-dependent nature of reward optimization (Booth et al., 2023). That is, whether a reward function is effective is influenced by the particular choice of RL algorithm, and the same reward may perform very differently even under the same optimizer given hyperparameter differences (Henderson et al., 2018; Agarwal et al., 2021). By providing detailed accounts on how well the RL algorithm optimizes individual reward components, reward reflection enables EUREKA to produce more targeted reward editing and reward functions that better synergize with the fixed RL algorithm.

## 4 EXPERIMENTS

We thoroughly evaluate EUREKA on a diverse suite of robot embodiments and tasks, testing its ability to generate reward functions, solve new tasks, and incorporate various forms of human input. We use GPT-4 (OpenAI, 2023), in particular the `gpt-4-0314` variant, as the backbone LLM for all LLM-based reward-design algorithms unless specified otherwise.

**Environments.** Our environments consist of 10 distinct robots and 29 tasks implemented using the IsaacGym simulator (Makoviychuk et al., 2021). First, we include 9 original environments from IsaacGym (**Isaac**), covering a diverse set of robot morphologies from quadruped, bipedal, quadrotor, cobot arm, to dexterous hands. In addition to coverage over robot form factors, we ensure *depth* in our evaluation by including all 20 tasks from the Bidexterous Manipulation (**Dexterity**) benchmark (Chen et al., 2022). Dexterity contains 20 complex bi-manual tasks that require a pair of Shadow Hands to solve a wide range of complex manipulation skills, ranging from object handover to rotating a cup by 180 degrees. For the task description input to EUREKA, we use the official description provided in the environment repository when possible. See App. C for details on all environments. It is worth noting that both benchmarks are publicly released concurrently, or after the GPT-4 knowledge cut-off date (September 2021), so GPT-4 is unlikely to have accumulated extensive internet knowledge about these tasks, making them ideal testbeds for assessing EUREKA's reward generation capability compared to measurable human-engineered reward functions.

### 4.1 BASELINES

**L2R** (Yu et al., 2023) proposes a two-stage LLM-prompting solution to generate templated rewards. For an environment and task specified in natural language, a first LLM is asked to fill in a natural language template describing the agent's motion; then, a second LLM is asked to convert this "motion description" into code that calls a manually defined set of reward API primitives to write a reward program that sets their parameters. To make L2R competitive for our tasks, we define the motion description template to mimic the original L2R templates, and we construct the API reward primitives using the individual components of the original human rewards when possible. Note that this gives L2R an advantage as it has access to the original reward functions. Consistent with EUREKA, we conduct 5 independent L2R runs per environment, and for each run, we generate 16 reward samples. See App. D for more details.

**Human.** These are the original shaped reward functions provided in our benchmark tasks. As these reward functions are written by active reinforcement learning researchers who designed the tasks, these reward functions represent the outcomes of expert-level human reward engineering.

**Sparse.** These are identical to the fitness functions $F$ that we use to evaluate the quality of the generated rewards. Like Human, these are also provided by the benchmark. On the dexterity tasks, they are uniformly binary indicator functions that measure task success; on Isaac tasks, they vary in functional forms depending on the nature of the task. See App. C for a description of the ground-truth scoring metric for all tasks.

### 4.2 TRAINING DETAILS

**Policy Learning.** For each task, all final reward functions are optimized using the same RL algorithm with the same set of hyperparameters. Isaac and Dexterity share a well-tuned PPO implementation (Schulman et al., 2017; Makoviichuk & Makoviychuk, 2021), and we use this implementation and the task-specific PPO hyperparameters without any modification. Note that these task hyperparameters are tuned to make the official human-engineered rewards work well. For each reward, we run 5 independent PPO training runs and report the average of the maximum task metric values achieved by policy checkpoints as the reward's performance.

**Reward Evaluation Metrics.** For Isaac tasks, since the task metric $F$ for each task varies in semantic meaning and scale, we report the **human normalized score** for EUREKA and L2R, $\frac{\text{Method} - \text{Sparse}}{|\text{Human} - \text{Sparse}|}$. This metric provides a holistic measure of how EUREKA rewards fare against human-expert rewards with respect to the ground-truth task metric. For Dexterity, since all tasks are evaluated using the binary success function, we directly report success rates.

### 4.3 RESULTS

**EUREKA outperforms human rewards.** In Figure 3, we report the aggregate results on Dexterity and Isaac. Notably, EUREKA exceeds or performs on par to human level on *all* Isaac tasks and 15 out of 20 tasks on Dexterity (see App. F for a per-task breakdown). In contrast, L2R, while comparable on low-dimensional tasks (e.g., CartPole, BallBalance), lags significantly behind on high-dimensional tasks. Despite being provided access to some of the same reward components as Human, L2R still underperforms EUREKA after its initial iteration, when both methods have had the same number of reward queries. As expected, L2R's lack of expressivity severely limits its performance. In contrast, EUREKA generates free-form rewards from scratch without any domain-specific knowledge and performs substantially better. In App. F, we ablate GPT-4 with GPT-3.5 and find EUREKA degrades in performance but still matches or exceeds human-level on most Isaac tasks, indicating that its general principles can be readily applied to coding LLMs of varying qualities.

**EUREKA consistently improves over time.** In Fig. 4, we visualize the average performance of the cumulative best EUREKA rewards after each evolution iteration. Moreover, we study an ablation, **EUREKA w.o. Evolution (32 Samples)**, which performs only the initial reward generation step, sampling the same number of reward functions as two iterations in the original EUREKA. This ablation helps study, given a fixed number of reward function budget, whether it is more advantageous to perform the

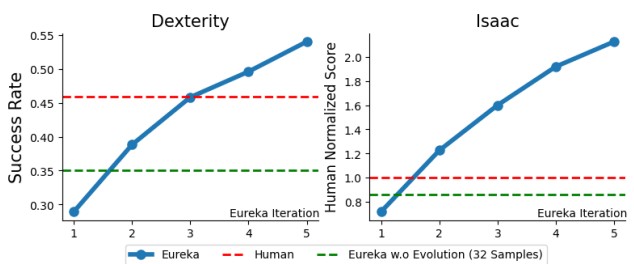

Figure 4: EUREKA progressively produces better rewards via in-context evolutionary reward search.

EUREKA evolution or simply sample more first-attempt rewards without iterative improvement. As seen, on both benchmarks, EUREKA rewards steadily improve and eventually surpass human rewards in performance despite sub-par initial performances. This consistent improvement also cannot be replaced by just sampling more in the first iteration as the ablation's performances are lower than EUREKA after 2 iterations on both benchmarks. Together, these results demonstrate that EUREKA's novel evolutionary optimization is indispensable for its final performance.

**EUREKA generates novel rewards.** We assess the novelty of EUREKA rewards by computing the *correlations* between EUREKA and human rewards on all the Isaac tasks; see App. C for details on this procedure. Then, we plot the correlations against the human normalized scores on a scatter-plot in Figure 5, where each point represents a single EUREKA reward on a single task. As shown, EUREKA mostly generates weakly correlated reward functions that outperform the human ones. In addition, by examining the average correlation by task (App. F), we observe that *the harder the task is, the less correlated the EUREKA rewards*. We hypothesize that human rewards are less likely to be near optimal for difficult tasks, leaving more room for EUREKA rewards to be different and better.

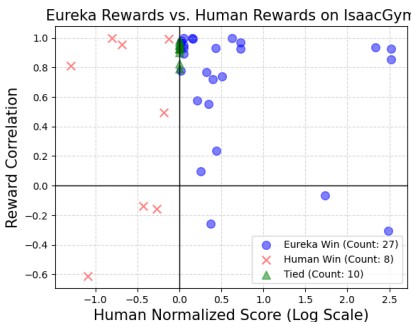

Figure 5: Eureka generates novel rewards.

In a few cases, EUREKA rewards are even *negatively* correlated with human rewards but perform significantly better, demonstrating that EUREKA can *discover* novel reward design principles that may run counter to human intuition; we illustrate these EUREKA rewards in App. G.2.

**Reward reflection enables targeted improvement.** To assess the importance of constructing reward reflection in the reward feedback, we evaluate an ablation, **EUREKA (No Reward Reflection)**, which reduces the reward feedback prompt to include only snapshot values of the task metric $F$. Averaged over all Isaac tasks, EUREKA without reward reflection reduces the average normalized score by 28.6%; in App. F, we provide detailed per-task breakdown and observe much greater performance deterioration on higher dimensional tasks. To provide qualitative analysis, in App. G.1, we include several examples in which EUREKA utilizes the reward reflection to perform targeted reward editing.

**EUREKA with curriculum learning enables dexterous pen spinning.** Finally, we investigate whether EUREKA can be used to solve a truly novel and challenging dexterous task. To this end, we propose pen spinning as a test bed. This task is highly dynamic and requires a Shadow Hand to continuously rotate a pen to achieve some predefined spinning patterns for as many cycles as possible. We consider a *curriculum learning* (Bengio et al., 2009) approach to break down the task into manageable components that can be independently solved by EUREKA; similar approaches have been found successful for other coding LLM applications to decision making (Wang et al., 2023a). Specifically, we first instruct EUREKA to generate a reward for re-orienting the pen to random target configurations. Then, using this pre-trained policy (**Pre-Trained**), we fine-tune it using the EUREKA reward

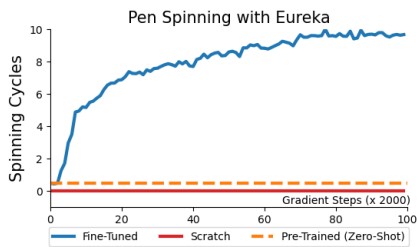

Figure 6: EUREKA can be flexibly combined with curriculum learning to acquire complex dexterous skills.

to reach the sequence of pen-spinning configurations (**Fine-Tuned**). To demonstrate the importance of curriculum learning, we also train a baseline policy from scratch using EUREKA reward without the first-stage pre-training (**Scratch**). The RL training curves are shown in Figure 6. Eureka fine-tuning quickly adapts the policy to successfully spin the pen for many cycles in a row; see Figure 1 and the supplementary video. In contrast, neither pre-trained or learning-from-scratch policies can complete even a single cycle of pen spinning. This result demonstrates EUREKA's applicability to advanced policy learning approaches that are often necessary for learning very complex skills. Using EUREKA to specify the *low-level* rewards, humans can simply focus on the high-level curriculum design and deliver a successful *co-pilot* solution to a challenging problem for the first time.

## 4.4 EUREKA FROM HUMAN FEEDBACK

In addition to automated reward design, EUREKA enables a new gradient-free in-context learning approach to RL from Human Feedback (RLHF) that can readily incorporate various types of human inputs to generate more performant and human-aligned reward functions.

**EUREKA can improve and benefit from human reward functions.** We study whether starting with a human reward function initialization, a common scenario in real-world RL applications, is advantageous for EUREKA. Importantly, incorporating human initialization requires no modification to

EUREKA – we can simply substitute the raw human reward function as the output of the first EUREKA iteration. To investigate this, we select several tasks from Dexterity that differ in the relative performances between the original EUREKA and human rewards. The full results are shown in Figure 7. As shown, regardless of the quality of the human rewards, EUREKA improves and benefits from human rewards as **EUREKA (Human Init.)** is uniformly better than both EUREKA and Human on all tasks.

This suggests that EUREKA's in-context reward improvement capability is largely independent of the quality of the base reward. Furthermore, the fact that EUREKA can significantly improve over human rewards even when they are highly sub-optimal hints towards an interesting hypothesis: *human designers are generally knowledgeable about relevant state variables but are less proficient at designing rewards using them.* This makes intuitive sense as identifying relevant state variables that should be included in the reward function involves mostly common sense reasoning, but reward design requires specialized knowledge and experience in RL. Together, these results demonstrate EUREKA's *reward assistant* capability, perfectly complementing human designers' knowledge about useful state variables and making up for their less proficiency on how to design rewards using them. In App. G.3, we provide several examples of EUREKA (Human Init.) steps.

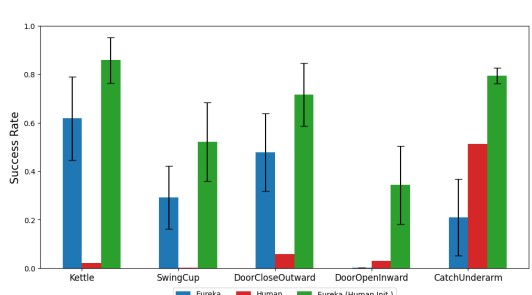

Figure 7: EUREKA effectively improves and benefits from human reward initialization.

**Reward reflection via human feedback induces aligned behavior.** So far, all EUREKA rewards are optimized against a fixed, black-box task fitness function $F$. This task metric, however, may not fully align with human intent. Moreover, in many open-ended tasks, $F$ may not be available in the first place (Fan et al., 2022). In these challenging scenarios, we propose to augment EUREKA by having humans step in and put into words the reward reflection in terms of the desired behavior and correction. We investigate this capability in EUREKA by teaching a Humanoid agent how to run purely from textual reward reflection; in App. G.4, we show the exact sequence of human feedback and EUREKA

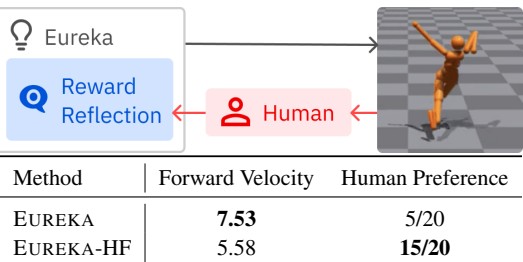

| Method | Forward Velocity | Human Preference |
|---|---|---|
| EUREKA | **7.53** | 5/20 |
| EUREKA-HF | 5.58 | **15/20** |

Figure 8: EUREKA can incorporate human reward reflection to modify rewards that induce safer and more human-aligned behavior.

rewards. Then, we conduct a user study asking 20 unfamiliar users to indicate their preferences between two policy rollout videos shown in random order, one trained with human reward reflection (**EUREKA-HF**) and the other one trained with the original best EUREKA reward; the details are in App. E.2. As shown in Fig. 8, despite running a bit slower, the EUREKA-HF agent is preferred by a large majority of our users. Qualitative, we indeed see that the EUREKA-HF agent acquires safer and more stable gait, as instructed by the human. See the supplementary videos for a comparison.

## 5 CONCLUSION

We have presented EUREKA, a universal reward design algorithm powered by coding large language models and in-context evolutionary search. Without any task-specific prompt engineering or human intervention, EUREKA achieves human-level reward generation on a wide range of robots and tasks. EUREKA's particular strength in learning dexterity solves dexterous pen spinning for the first time with a curriculum learning approach. Finally, EUREKA enables a gradient-free approach to reinforcement learning from human feedback that readily incorporates human reward initialization and textual feedback to better steer its reward generation. The versatility and substantial performance gains of EUREKA suggest that the simple principle of combining large language models with evolutionary algorithms are a general and scalable approach to reward design, an insight that may be generally applicable to difficult, open-ended search problems.

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

## A  RELATED WORK

**Reward Design.** Reward engineering is a long-standing challenge in reinforcement learning (Singh et al., 2010; Sutton & Barto, 2018). The most common reward design method is manual trial-and-error (Knox et al., 2023; Booth et al., 2023). Inverse reinforcement learning (IRL) infers reward functions from demonstrations (Abbeel & Ng, 2004; Ziebart et al., 2008; Ho & Ermon, 2016), but it requires expensive expert data collection, which may not be available, and outputs non-interpretable black-box reward functions. Several prior works have studied automated reward search through evolutionary algorithms (Niekum et al., 2010; Chiang et al., 2019; Faust et al., 2019). These early attempts are limited to task-specific implementations of evolutionary algorithms that search over only parameters within provided reward templates. Recent works have also proposed using pretrained foundation models that can produce reward functions for new tasks (Ma et al., 2022; 2023; Fan et al., 2022; Du et al., 2023a; Karamcheti et al., 2023; Du et al., 2023b; Kwon et al., 2023). Most of these approaches output *scalar* rewards that lack interpretability and do not naturally admit the capability to improve or adapt rewards on-the-fly. In contrast, EUREKA adeptly generates free-form, white-box reward code and effectively in-context improves.

**Code Large Language Models for Decision Making.** Recent works have considered using coding LLMs (Austin et al., 2021; Chen et al., 2021; Rozière et al., 2023) to generate grounded and structured programmatic output for decision making and robotics problems (Liang et al., 2023; Singh et al., 2023; Wang et al., 2023b; Huang et al., 2023; Wang et al., 2023a; Liu et al., 2023; Silver et al., 2023; Ding et al., 2023; Lin et al., 2023; Xie et al., 2023). However, most of these works rely on known motion primitives to carry out robot actions and do not apply to robot tasks that require low-level skill learning, such as dexterous manipulation. The closest to our work is a recent work (Yu et al., 2023) that also explores using LLMs to aid reward design. Their approach, however, requires domain-specific task descriptions and reward templates, which demand substantial domain knowledge and limit the expressivity of the generated reward functions.

**Evolution with LLMs.** Implementing evolutionary algorithms with LLMs has been explored in recent works in the context of neural architecture search (Chen et al., 2023; Nasir et al., 2023), prompt engineering (Guo et al., 2023), as well as morphology design (Lehman et al., 2022). Ours is the first to apply this principle to reward design. Unlike prior approaches, EUREKA does not need humans to provide the initial candidates or few-shot prompting. Furthermore, EUREKA introduces novel reward reflection mechanism that enables more targeted and effective reward mutation.

## B  FULL PROMPTS

In this section, we provide all EUREKA prompts.

Prompt 1: Initial system prompt

```
You are a reward engineer trying to write reward functions to solve reinforcement learning
    tasks as effective as possible.
Your goal is to write a reward function for the environment that will help the agent learn the
    task described in text.
Your reward function should use useful variables from the environment as inputs. As an example
    ,
the reward function signature can be:
@torch.jit.script
def compute_reward(object_pos: torch.Tensor, goal_pos: torch.Tensor) -> Tuple[torch.Tensor,
    Dict[str, torch.Tensor]]:
    ...
    return reward, {}
Since the reward function will be decorated with @torch.jit.script,
please make sure that the code is compatible with TorchScript (e.g., use torch tensor instead
    of numpy array).
Make sure any new tensor or variable you introduce is on the same device as the input tensors.
```

Prompt 2: Reward reflection and feedback

```
We trained a RL policy using the provided reward function code and tracked the values of the
    individual components in the reward function as well as global policy metrics such as
    success rates and episode lengths after every {epoch_freq} epochs and the maximum, mean,
    minimum values encountered:
<REWARD REFLECTION HERE>
```

```
Please carefully analyze the policy feedback and provide a new, improved reward function that
    can better solve the task. Some helpful tips for analyzing the policy feedback:
    (1) If the success rates are always near zero, then you must rewrite the entire reward
    function
    (2) If the values for a certain reward component are near identical throughout, then this
    means RL is not able to optimize this component as it is written. You may consider
        (a) Changing its scale or the value of its temperature parameter
        (b) Re-writing the reward component
        (c) Discarding the reward component
    (3) If some reward components' magnitude is significantly larger, then you must re-scale
    its value to a proper range
Please analyze each existing reward component in the suggested manner above first, and then
    write the reward function code.
```

Prompt 3: Code formatting tip

```
The output of the reward function should consist of two items:
    (1) the total reward,
    (2) a dictionary of each individual reward component.
The code output should be formatted as a python code string: "```python ... ```".

Some helpful tips for writing the reward function code:
    (1) You may find it helpful to normalize the reward to a fixed range by applying
    transformations like torch.exp to the overall reward or its components
    (2) If you choose to transform a reward component, then you must also introduce a
    temperature parameter inside the transformation function; this parameter must be a named
    variable in the reward function and it must not be an input variable. Each transformed
    reward component should have its own temperature variable
    (3) Make sure the type of each input variable is correctly specified; a float input
    variable should not be specified as torch.Tensor
    (4) Most importantly, the reward code's input variables must contain only attributes of
    the provided environment class definition (namely, variables that have prefix self.).
    Under no circumstance can you introduce new input variables.
```

## C    ENVIRONMENT DETAILS

In this section, we provide environment details. For each environment, we list its observation and action dimensions, the verbatim task description, and the task fitness function $F$. $F$ is evaluated per-environment, and our policy feedback uses the mean across environment instances.

For the functions below, `||` denotes the $L_2$ norm, and `1[]` denotes the indicator function.

| IsaacGym Environments |
|---|
| Environment (obs dim, action dim) |
| Task description |
| Task fitness function $F$ |
| Cartpole (4, 1) |
| To balance a pole on a cart so that the pole stays upright |
| `duration` |
| Quadcopter (21, 12) |
| To make the quadcopter reach and hover near a fixed position |
| `-cur_dist` |
| FrankaCabinet (23, 9) |
| To open the cabinet door |
| `1[cabinet_pos > 0.39]` |
| Anymal (48, 12) |
| To make the quadruped follow randomly chosen x, y, and yaw target velocities |
| `-(linvel_error + angvel_error)` |
| BallBalance (48, 12) |
| To keep the ball on the table top without falling |

| duration |
| --- |
| Ant (60, 8) |
| To make the ant run forward as fast as possible |
| `cur_dist - prev_dist` |
| AllegroHand (88, 16) |
| To make the hand spin the object to a target orientation |
| `number of consecutive successes where`
`current success is 1[rot_dist < 0.1]` |
| Humanoid (108, 21) |
| To make the humanoid run as fast as possible |
| `cur_dist - prev_dist` |
| ShadowHand (211, 20) |
| To make the shadow hand spin the object to a target orientation |
| `number of consecutive successes where`
`current success is 1[rot_dist < 0.1]` |

| Dexterity Environments |
| --- |
| Environment (obs dim, action dim) |
| Task description |
| Task fitness function $F$ |
| Over (398, 40) |
| This class corresponds to the HandOver task. This environment consists of two shadow hands with palms facing up, opposite each other, and an object that needs to be passed. In the beginning, the object will fall randomly in the area of the shadow hand on the right side. Then the hand holds the object and passes the object to the other hand. Note that the base of the hand is fixed. More importantly, the hand which holds the object initially can not directly touch the target, nor can it directly roll the object to the other hand, so the object must be thrown up and stays in the air in the process |
| `1[dist < 0.03]` |
| DoorCloseInward (417, 52) |
| This class corresponds to the DoorCloseInward task. This environment require a closed door to be opened and the door can only be pushed outward or initially open inward. Both these two environments only need to do the push behavior, so it is relatively simple |
| `1[door_handle_dist < 0.5]` |
| DoorCloseOutward (417, 52) |
| This class corresponds to the DoorCloseOutward task. This environment also require a closed door to be opened and the door can only be pushed inward or initially open outward, but because they can't complete the task by simply pushing, which need to catch the handle by hand and then open or close it, so it is relatively difficult |
| `1[door_handle_dist < 0.5]` |
| DoorOpenInward (417, 52) |
| This class corresponds to the DoorOpenInward task. This environment also require a opened door to be closed and the door can only be pushed inward or initially open outward, but because they can't complete the task by simply pushing, which need to catch the handle by hand and then open or close it, so it is relatively difficult |
| `1[door_handle_dist > 0.5]` |
| DoorOpenOutward (417, 52) |
| This class corresponds to the DoorOpenOutward task. This environment require a opened door to be closed and the door can only be pushed outward or initially open inward. Both these two environments only need to do the push behavior, so it is relatively simple |

```
1[door_handle_dist < 0.5]
```

**Scissors (417, 52)**

This class corresponds to the Scissors task. This environment involves two hands and scissors, we need to use two hands to open the scissors

```
1[dof_pos > -0.3]
```

**SwingCup (417, 52)**

This class corresponds to the SwingCup task. This environment involves two hands and a dual handle cup, we need to use two hands to hold and swing the cup together

```
1[rot_dist < 0.785]
```

**Switch (417, 52)**

This class corresponds to the Switch task. This environment involves dual hands and a bottle, we need to use dual hand fingers to press the desired button

```
1[1.4 - (left_switch_z + right_switch_z) > 0.05]
```

**Kettle (417, 52)**

This class corresponds to the PourWater task. This environment involves two hands, a kettle, and a bucket, we need to hold the kettle with one hand and the bucket with the other hand, and pour the water from the kettle into the bucket. In the practice task in Isaac Gym, we use many small balls to simulate the water

```
1[|bucket - kettle_spout| < 0.05]
```

**LiftUnderarm (417, 52)**

This class corresponds to the LiftUnderarm task. This environment requires grasping the pot handle with two hands and lifting the pot to the designated position. This environment is designed to simulate the scene of lift in daily life and is a practical skill

```
1[dist < 0.05]
```

**Pen (417, 52)**

This class corresponds to the Open Pen Cap task. This environment involves two hands and a pen, we need to use two hands to open the pen cap

```
1[5 * |pen_cap - pen_body| > 1.5]
```

**BottleCap (420, 52)**

This class corresponds to the Bottle Cap task. This environment involves two hands and a bottle, we need to hold the bottle with one hand and open the bottle cap with the other hand. This skill requires the cooperation of two hands to ensure that the cap does not fall

```
1[dist > 0.03]
```

**CatchAbreast (422, 52)**

This class corresponds to the Catch Abreast task. This environment consists of two shadow hands placed side by side in the same direction and an object that needs to be passed. Compared with the previous environment which is more like passing objects between the hands of two people, this environment is designed to simulate the two hands of the same person passing objects, so different catch techniques are also required and require more hand translation and rotation techniques

```
1[dist] < 0.03
```

**CatchOver2Underarm (422, 52)**

This class corresponds to the Over2Underarm task. This environment is similar to Catch Underarm, but with an object in each hand and the corresponding goal on the other hand. Therefore, the environment requires two objects to be thrown into the other hand at the same time, which requires a higher manipulation technique than the environment of a single object

```
1[dist < 0.03]
```

**CatchUnderarm (422, 52)**

This class corresponds to the Catch Underarm task. In this task, two shadow hands with palms facing upwards are controlled to pass an object from one palm to the other. What makes it more difficult than the Hand over task is that the hands' translation and rotation degrees of freedom are no longer frozen but are added into the action space

```
1[dist < 0.03]
```

ReOrientation (422, 40)

This class corresponds to the ReOrientation task. This environment involves two hands and two objects. Each hand holds an object and we need to reorient the object to the target orientation

```
1[rot_dist < 0.1]
```

GraspAndPlace (425, 52)

This class corresponds to the GraspAndPlace task. This environment consists of dual-hands, an object and a bucket that requires us to pick up the object and put it into the bucket

```
1[|block - bucket| < 0.2]
```

BlockStack (428, 52)

This class corresponds to the Block Stack task. This environment involves dual hands and two blocks, and we need to stack the block as a tower

```
1[goal_dist_1 < 0.07 and goal_dist_2 < 0.07 and
50 * (0.05 - z_dist_1) > 1]
```

PushBlock (428, 52)

This class corresponds to the PushBlock task. This environment involves two hands and two blocks, we need to use both hands to reach and push the block to the desired goal separately. This is a relatively simple task

```
1[left_dist <= 0.1 and right_dist <= 0.1] +
0.5 * 1[left_dist <= 0.1 and right_dist > 0.1]
```

TwoCatchUnderarm (446, 52)

This class corresponds to the TwoCatchUnderarm task. This environment is similar to Catch Underarm, but with an object in each hand and the corresponding goal on the other hand. Therefore, the environment requires two objects to be thrown into the other hand at the same time, which requires a higher manipulation technique than the environment of a single object

```
1[goal_dist_1 + goal_dist_2 < 0.06]
```

## D  BASELINE DETAILS

Language-to-Rewards (L2R) uses an LLM to generate a motion description from a natural language instruction and a set of reward API calls from the motion description. The reward is computed as the sum of outputs from the reward API calls. While the LLM automates the process of breaking down the task into basic low-level instructions, manual effort is still required to specify the motion description template, low-level reward API, and the API's function implementations.

All three parts require significant design considerations and can drastically affect L2R's performance and capabilities. Unfortunately, this makes comparison difficult since L2R requires manual engineering whereas Eureka is fully automatic—ambiguity thus arises from how much human-tuning should be done with L2R's components. Nonetheless, we seek to provide a fair comparison and base our implementation off two factors:

- To create our motion description template, we reference L2R's quadruped and dexterous manipulator templates. Specifically, our templates consist of statements that set parameters to quantitative values and statements that relate two parameters. We also aim to mimic the style of L2R's template statements in general.

- The reward API is designed so that each template statement can be faithfully written in terms of an API function. The functions are implemented to resemble their respective human reward terms from their environment; thus, L2R is given an advantage in that its components resemble the manually-tuned human reward. In a few exceptions where the human reward differs significantly from the L2R template style, we base our API implementation on the formulas provided in the L2R appendix.

L2R was designed to allow for an agent in a single environment to perform multiple tasks. Thus, each environment has its own motion description template and reward API. Since our experiments

range over many agents and environments, we have one template and API for each Isaac task, and we generalize all Dexterity tasks into one environment with all necessary objects.

For illustration, our descriptor and coder prompts for the Dexterity experiments are below.

Prompt 1: Dexterity descriptor prompt

```
We have two dexterous manipulators (shadow hands) and we want you to help plan how it should
    move to perform tasks using the following template:

[start of description]
object1={CHOICE: <INSERT OBJECTS HERE>} should be {CHOICE: close to, far from} object2={CHOICE
    : <INSERT OBJECTS HERE>, nothing}.
[optional] object3={CHOICE: <INSERT OBJECTS HERE>} should be {CHOICE: close to, far from}
    object4={CHOICE: <INSERT OBJECTS HERE>, nothing}.
[optional] object1 needs to have a rotation orientation similar to object2.
[optional] object3 needs to have a rotation orientation similar to object4.
<INSERT OPTIONAL HAND DESCRIPTIONS HERE>
[optional] doors needs to be {CHOICE: open, closed} {CHOICE: inward, outward}.
[optional] scissor needs to be opened to [NUM: 0.0] radians.
[optional] block2 needs to be stacked on top of block1.
[end of description]

Rules:
1. If you see phrases like [NUM: default_value], replace the entire phrase with a numerical
    value.
2. If you see phrases like {CHOICE: choice1, choice2, ...}, it means you should replace the
    entire
phrase with one of the choices listed.
3. If you see [optional], it means you only add that line if necessary for the task, otherwise
    remove that line.
4. The environment contains <INSERT OBJECTS HERE>. Do not invent new objects not listed here.
5. I will tell you a behavior/skill/task that I want the manipulator to perform and you will
    provide the full plan, even if you may only need to change a few lines. Always start the
    description with [start of plan] and end it with [end of plan].
6. You can assume that the hands are capable of doing anything, even for the most challenging
    task.
7. Your plan should be as close to the provided template as possible. Do not include
    additional details.
```

Prompt 2: Dexterity coder prompt

```
We have a plan of a robot arm with palm to manipulate objects and we want you to turn that
    into the corresponding program with following functions:

```
def set_min_l2_distance_reward(name_obj_A, name_obj_B)
```
This term sets a reward for minimizing l2 distance between name_obj_A and name_obj_B so they
    get closer to each other.
name_obj_A and name_obj_B are selected from [<INSERT FIELDS HERE>].

```
def set_max_l2_distance_reward(name_obj_A, name_obj_B)
```
This term sets a reward for maximizing l2 distance between name_obj_A and name_obj_B so they
    get closer to each other.
name_obj_A and name_obj_B are selected from [<INSERT FIELDS HERE>].

```
def set_obj_orientation_reward(name_obj_A, name_obj_B)
```
This term encourages the orientation of name_obj_A to be close to the orientation of
    name_obj_B. name_obj_A and name_obj_B are selected from [<INSERT ORIENTATION FIELDS HERE
    >].

Example plan:
object1=object1 should be close to object2=object1_goal.
object1 needs to have a rotation orientation similar to object2.
To perform this task, the left manipulator's palm should move close to object1.

Example answer code:
```
set_min_l2_distance_reward("object1", "object1_goal")
set_min_l2_distance_reward("object1", "left_palm")
set_obj_orientation_reward("object1", "object1_goal")
```

Remember:
```

```
1. Always format the code in code blocks.
2. Do not wrap your code in a function. Your output should only consist of function calls like
     the example above.
3. Do not invent new functions or classes. The only allowed functions you can call are the
     ones listed above, and do not implement them. Do not leave unimplemented code blocks in
     your response.
4. The only allowed library is numpy. Do not import or use any other library.
5. If you are not sure what value to use, just use your best judge. Do not use None for
     anything.
6. Do not calculate the position or direction of any object (except for the ones provided
     above). Just use a number directly based on your best guess.
7. You do not need to make the robot do extra things not mentioned in the plan such as
     stopping the robot.
```

For the sections surrounded by angle brackets <>, we specify a list of valid objects for each Dexterity task. For example, ShadowHandPen's list of objects is defined as follows:

```
"shadow_hand_pen": ["left_palm", "right_palm", "left_forefinger", "left_middlefinger", "
    left_ringfinger", "left_littlefinger", "left_thumb", "right_forefinger", "
    right_middlefinger", "right_ringfinger", "right_littlefinger", "right_thumb", "pen_cap",
    "pen"]
```

A summary of terms and their implementations for each experiment is in Table 3. Note that many environments automatically randomize their target parameters during training after a reset or success criteria is met, which L2R cannot account for during the reward generation stage. Thus, while L2R's experiments define targets in terms of quantitative values, it's incompatible with our environments, and we define targets instead as relations between two parameters (usually the object and the object's target).

## D.1 L2R REWARD EXAMPLES

Example 1: L2R reward function on Humanoid, Human Normalized Score: 0.0

```
set_torso_height_reward(1.1)
set_torso_velocity_reward(3.6)
set_angle_to_target_reward(0.0)
```

Example 2: L2R reward function on ShadowHandKettle, Success Rate: 0.07

```
set_min_l2_distance_reward("kettle_handle", "bucket_handle")
set_min_l2_distance_reward("kettle_spout", "bucket_handle")
set_min_l2_distance_reward("left_palm", "bucket_handle")
set_min_l2_distance_reward("right_palm", "kettle_handle")
set_min_l2_distance_reward("left_thumb", "bucket_handle")
set_min_l2_distance_reward("right_thumb", "kettle_handle")
```

## E EUREKA DETAILS

**Environment as Context.** In Isaac Gym, the simulator adopts a environment design pattern in which the environment observation code is typically written inside a `compute_observations()` function within the environment object class; this applies to all our environments. Therefore, we have written an automatic script to extract just the observation portion of the environment source code. This is done largely to reduce our experiment cost as longer context induces higher cost. Furthermore, given that current LLMs have context length limit, this task agnostic way of trimming the environment code before feeing it to the context allows us to fit every environment source code into context.

Example 1: Humanoid environment observation given to EUREKA.

```
class Humanoid(VecTask):
    """Rest of the environment definition omitted."""
    def compute_observations(self):
        self.gym.refresh_dof_state_tensor(self.sim)
        self.gym.refresh_actor_root_state_tensor(self.sim)

        self.gym.refresh_force_sensor_tensor(self.sim)
        self.gym.refresh_dof_force_tensor(self.sim)
        self.obs_buf[:], self.potentials[:], self.prev_potentials[:], self.up_vec[:], self.
    heading_vec[:] = compute_humanoid_observations(
```

| Reward Term | Formulation |
| --- | --- |
| **Dexterity** | |
| Minimize distance | $-\|p_1 - p_2\|_2$ |
| Maximize distance | $\|p_1 - p_2\|_2$ |
| Minimize orientation | $2\arcsin(\min(\|v(q_1\bar{q}_2)\|_2, 1))$ |
| **AllegroHand** | |
| Minimize distance | $-\|p_1 - p_2\|_2$ |
| Maximize distance | $\|p_1 - p_2\|_2$ |
| Minimize orientation difference | $1/(|2\arcsin(\min(\|v(q_1\bar{q}_2)\|_2, 1))| + \epsilon)$ |
| Maximize orientation difference | $-1/(|2\arcsin(\min(\|v(q_1\bar{q}_2)\|_2, 1))| + \epsilon)$ |
| **Ant** | |
| Torso height | $-|h - h_t|$ |
| Torso velocity | $-|\|v_{xy}\|_2 - v_t|$ |
| Angle to target | $-|\theta - \theta_t|$ |
| **Anymal** | |
| Minimize difference | $\exp -(x - x_t)^2$ |
| **BallBalance** | |
| Ball position | $1/(1 + \|p - p_t\|_2)$ |
| Ball velocity | $1/(1 + \|v - v_t\|_2)$ |
| **Cartpole** | |
| Pole angle | $-(\theta - \theta_t)^2$ |
| Pole velocity | $-|v - v_t|$ |
| Cart velocity | $-|v - v_t|$ |
| **FrankaCabinet** | |
| Minimize hand distance | $-\|p_1 - p_2\|_2$ |
| Maximize hand distance | $\|p_1 - p_2\|_2$ |
| Drawer extension | $-|p - p_t|$ |
| **Humanoid** | |
| Torso height | $-|h - h_t|$ |
| Torso velocity | $-|\|v_{xy}\|_2 - v_t|$ |
| Angle to target | $-|\theta - \theta_t|$ |
| **Quadcopter** | |
| Quadcopter position | $1/(1 + \|p - p_t\|_2^2)$ |
| Upright alignment | $1/(1 + |1 - n_z|^2)$ |
| Positional velocity | $1/(1 + \|v - v_t\|_2^2)$ |
| Angular velocity | $1/(1 + \|\omega - \omega_t\|_2^2)$ |
| **ShadowHand** | |
| Minimize distance | $-\|p_1 - p_2\|_2$ |
| Maximize distance | $\|p_1 - p_2\|_2$ |
| Minimize orientation difference | $1/(|2\arcsin(\min(\|v(q_1\bar{q}_2)\|_2, 1))| + \epsilon)$ |
| Maximize orientation difference | $-1/(|2\arcsin(\min(\|v(q_1\bar{q}_2)\|_2, 1))| + \epsilon)$ |

Table 3: L2R reward primitives and their implementations. $v(q)$ denotes the vector part of quaternion $q$, subscript $t$ denotes target value, and $n$ denotes the normal vector (orientation). All components are weighed equally.

```
        self.obs_buf, self.root_states, self.targets, self.potentials,
        self.inv_start_rot, self.dof_pos, self.dof_vel, self.dof_force_tensor,
        self.dof_limits_lower, self.dof_limits_upper, self.dof_vel_scale,
        self.vec_sensor_tensor, self.actions, self.dt, self.contact_force_scale, self.
angular_velocity_scale,
        self.basis_vec0, self.basis_vec1)
```

```
def compute_humanoid_observations(obs_buf, root_states, targets, potentials, inv_start_rot,
    dof_pos, dof_vel,
                                  dof_force, dof_limits_lower, dof_limits_upper, dof_vel_scale
    ,
                                  sensor_force_torques, actions, dt, contact_force_scale,
    angular_velocity_scale,
                                  basis_vec0, basis_vec1):
    # type: (Tensor, Tensor, Tensor, Tensor, Tensor, Tensor, Tensor, Tensor, Tensor, Tensor,
     float, Tensor, Tensor, float, float, float, Tensor, Tensor) -> Tuple[Tensor, Tensor,
     Tensor, Tensor, Tensor]

    torso_position = root_states[:, 0:3]
    torso_rotation = root_states[:, 3:7]
    velocity = root_states[:, 7:10]
    ang_velocity = root_states[:, 10:13]

    to_target = targets - torso_position
    to_target[:, 2] = 0

    prev_potentials_new = potentials.clone()
    potentials = -torch.norm(to_target, p=2, dim=-1) / dt

    torso_quat, up_proj, heading_proj, up_vec, heading_vec = compute_heading_and_up(
        torso_rotation, inv_start_rot, to_target, basis_vec0, basis_vec1, 2)

    vel_loc, angvel_loc, roll, pitch, yaw, angle_to_target = compute_rot(
        torso_quat, velocity, ang_velocity, targets, torso_position)

    roll = normalize_angle(roll).unsqueeze(-1)
    yaw = normalize_angle(yaw).unsqueeze(-1)
    angle_to_target = normalize_angle(angle_to_target).unsqueeze(-1)
    dof_pos_scaled = unscale(dof_pos, dof_limits_lower, dof_limits_upper)

    obs = torch.cat((torso_position[:, 2].view(-1, 1), vel_loc, angvel_loc *
     angular_velocity_scale,
                     yaw, roll, angle_to_target, up_proj.unsqueeze(-1), heading_proj.unsqueeze
    (-1),
                     dof_pos_scaled, dof_vel * dof_vel_scale, dof_force * contact_force_scale,
                     sensor_force_torques.view(-1, 12) * contact_force_scale, actions), dim
    =-1)

    return obs, potentials, prev_potentials_new, up_vec, heading_vec
```

**EUREKA Reward History.** Given that LLMs have limited context, we also trim the EUREKA dialogue such that only the last reward and its reward reflection (in addition to initial system prompt) is kept in the context for the generation of the next reward. In other word, the reward improvement is *Markovian*. This is standard in gradient-free optimization, and we find this simplification to work well in practice.

**EUREKA Reward Evaluation.** All intermediate EUREKA reward functions are evaluated using 1 PPO run with the default task parameters. The final EUREKA reward, like all other baseline reward functions, are evaluated using 5 PPO runs with the average performance on the task fitness function $F$ as the reward performance.

**Human Normalized Score Reporting.** Given that there are several significant outliers in human normalized score when EUREKA is substantially better both Human and Sparse on a task, when reporting the average normalized improvement in our abstract, we adjust the score so that the normalized score must lie between $[0, 3]$ per task before computing the average over all 29 tasks.

## E.1 EUREKA FROM HUMAN INITIALIZATION

In our human initialization experiments, we use Eureka to improve human-written reward functions. This can be done by modifying the first EUREKA iteration to use the human reward in place of the LLM-generated one, thereby "assuming" that Eureka's first proposed reward is the human reward. To complete this iteration, we use the human reward in IsaacGym, compute feedback, and query the LLM to generate new reward functions based on the human reward and the reward reflections. Future iterations are identical to the default EUREKA setting.

To provide the human reward in the first iteration, we refactor the code slightly to be consistent with the EUREKA reward format, which exposes the individual reward components in a dictionary.

Furthermore, as human reward functions are often written in less interpretable fashion than EUREKA rewards (see App. G.5 for an example), we also strip away excess variables and parameters besides those needed for the actual reward computation.

### E.2    EUREKA FROM HUMAN FEEDBACK

In our human reward reflection experiment, we investigate whether humans can provide reward reflection for desideratum such as "running with natural gait" that may be difficult to express via a task fitness function. We repeat the EUREKA procedure with the following modifications: (1) We only sample 1 reward per iteration, and (2) a human textual input will replace the automatically constructed reward reflection as in the main experiment. To ensure that the human textual input do not require domain expertise, we have used feedback that is as colloquial as possible; the full conversation is shown in App. G.4.

After the EUREKA-HF agent is trained, we have asked 20 unfamiliar users to indicate their preferences between two videos shown in random order, one depicting the EUREKA-HF Humanoid agent and the other one depicting the original best EUREKA agent. These 20 users are other graduate and undergraduate students that have a wide range of familiarity in reinforcement learning, but are not involved with this research.

## F    ADDITIONAL RESULTS

**Dexterity Performance Breakdown.** We present the raw success rates of EUREKA, L2R, Human, and Sparse in Fig. 9.

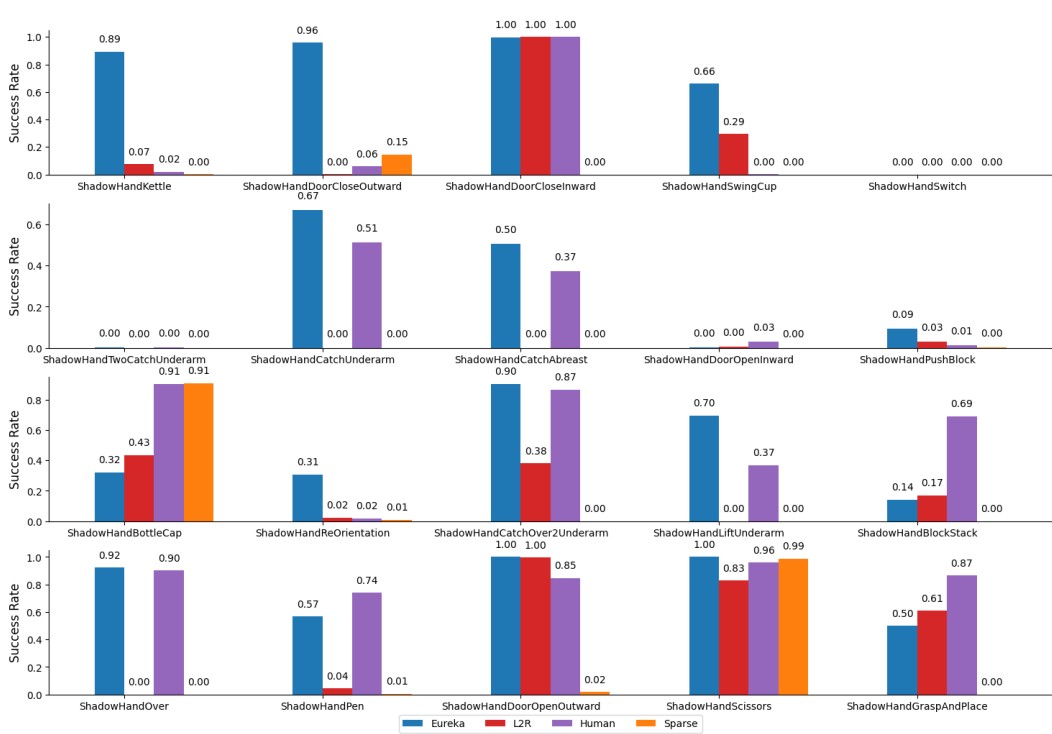

Figure 9: EUREKA rewards are less correlated with human rewards when the tasks are more high-dimensional and less common in the reinforcement learning literature.

**Reward Reflection Ablations.** In Fig. 10, we provide a detailed per-task breakdown on the impact of removing reward reflection in the EUREKA feedback. In this ablation, we are interested in the *average* human normalized score over independent EUREKA restarts because the average is more informative than the max (the metric used in all other experiments) in revealing LLM behavior change on aggregate. As shown, removing reward reflection generally has a negative impact on the reward performance. The deterioration is more pronounced for high-dimensional tasks, demonstrating that

reward reflection indeed can provide targeted reward editing that is more instrumental for difficult tasks that require many state components to interact in the reward functions.

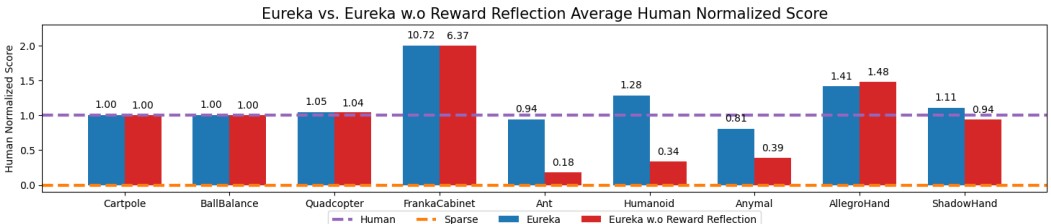

Figure 10: EUREKA rewards are less correlated with human rewards when the tasks are more high-dimensional and less common in the reinforcement learning literature.

**EUREKA with GPT-3.5.** In Fig. 11, we compare the performance of EUREKA with GPT-4 (the original one reported in the paper) and EUREKA with GPT-3.5; specifically, we use `gpt-3.5-turbo-16k-0613` in the OpenAI API. While the absolute performance goes down, we see that EUREKA (GPT-3.5) still performs comparably and exceeds human-engineered rewards on the dexterous manipulation tasks. These results suggest that the EUREKA principles are general and can be also applied to less performant base coding LLMs.

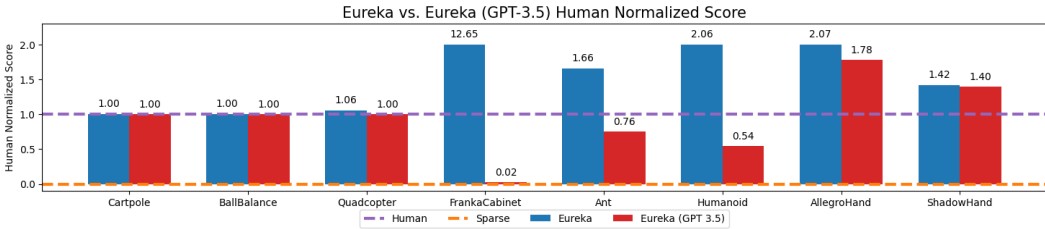

Figure 11: Using GPT3.5 observes performance degradation in EUREKA but still remains comparable to GPT-4 on a majority of the tasks.

**Reward Correlation Experiments.** To provide a more bird-eye view comparison against human rewards, we assess the novelty of EUREKA rewards. Given that programs that syntactically differ may functionally be identical, we propose to evaluate the Pearson *correlation* between EUREKA and human rewards on all the Isaac task. These tasks are ideal for this test because many of them have been widely used in RL research, even if the IsaacGym implementation may not have been seen in GPT-4 training, so it is possible that EUREKA produces rewards that are merely cosmetically different. To do this, for a given policy training run using a EUREKA reward, we gather all training transitions and compute their respectively EUREKA and human reward values, which can then be used to compute their correlation. Then, we plot the correlation against the human normalized score on a scatter-plot. The resulting scatter-plot is displayed in Fig. 5. In Fig. 12, we also provide the average correlation per task. As shown, as the tasks become more high-dimensional and harder to solve, the correlations exhibit a downward trend. This validates our hypothesis that the harder the task is, the less optimal the human rewards are, and consequently more room for EUREKA to generate truly novel and different rewards.

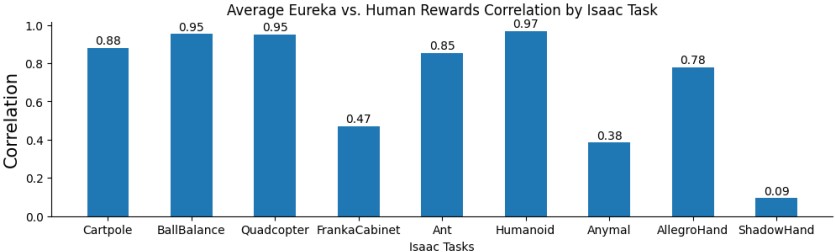

Figure 12: EUREKA rewards are less correlated with human rewards when the tasks are more high-dimensional and less common in the reinforcement learning literature.

```python
def compute_reward(object_rot, goal_rot, object_angvel, object_pos, fingertip_pos):
    # Rotation reward
    rot_diff = torch.abs(torch.sum(object_rot * goal_rot, dim=1) - 1) / 2
-   rotation_reward_temp = 20.0
+   rotation_reward_temp = 30.0                                              Changing hyperparameter
    rotation_reward = torch.exp(-rotation_reward_temp * rot_diff)

    # Distance reward
+   min_distance_temp = 10.0
    min_distance = torch.min(torch.norm(fingertip_pos - object_pos[:, None], dim=2), dim=1).values
-   distance_reward = min_distance
+   uncapped_distance_reward = torch.exp(-min_distance_temp * min_distance)
+   distance_reward = torch.clamp(uncapped_distance_reward, 0.0, 1.0)       Changing functional form

-   total_reward = rotation_reward + distance_reward
+   # Angular velocity penalty                                              Adding new component
+   angvel_norm = torch.norm(object_angvel, dim=1)
+   angvel_threshold = 0.5
+   angvel_penalty_temp = 5.0
+   angular_velocity_penalty = torch.where(angvel_norm > angvel_threshold,
+       torch.exp(-angvel_penalty_temp * (angvel_norm - angvel_threshold)), torch.zeros_like(angvel_norm))
+
+   total_reward = 0.5 * rotation_reward + 0.3 * distance_reward - 0.2 * angular_velocity_penalty

    reward_components = {
        "rotation_reward": rotation_reward,
        "distance_reward": distance_reward,
+       "angular_velocity_penalty": angular_velocity_penalty,
    }

    return total_reward, reward_components
```

Figure 13: EUREKA can zero-shot generate executable rewards and then flexibly improve them with many distinct types of free-form modification, such as (1) changing the hyperparameter of existing reward components, (2) changing the functional form of existing reward components, and (3) introducing new reward components.

# G  EUREKA REWARD EXAMPLES

In this section, we provide several unmodified EUREKA reward examples from various experiments we detailed in the main paper.

## G.1  REWARD REFLECTION EXAMPLES.

We illustrate EUREKA reward reflection step by displaying the sequence of previous EUREKA reward, EUREKA reward reflection, and then the subsequent EUREKA reward.

Example 1: EUREKA Reward Reflection on ShadowHand. Iteration 1 Score: 9.29, Iteration 2 Score: 10.43

```
Iteration 1: Eureka Output:
'''python
import torch
from typing import Tuple, Dict

@torch.jit.script
def compute_reward(object_rot: torch.Tensor, goal_rot: torch.Tensor) -> Tuple[torch.Tensor,
    Dict[str, torch.Tensor]]:
    rot_diff = torch.abs(torch.sum(object_rot * goal_rot, dim=1) - 1) / 2
    rotation_reward = torch.exp(-20 * rot_diff)

    # Scaling factor for the rotation reward
    rotation_temp = 20.0

    total_reward = rotation_reward

    reward_components = {
        "rotation_reward": rotation_reward
    }

    return total_reward, reward_components
'''
This reward function computes the difference between the current object rotation and the
    target rotation. It calculates a rotation_reward based on the absolute difference, scaled
     by a temperature parameter (rotation_temp). The total reward is the rotation_reward.

###################################################################
```

```
Iteration 1: Reward Reflection:
We trained a RL policy using the provided reward function code and tracked the values of the
    individual components in the reward function as well as global policy metrics such as
    success rates and episode lengths after every 300 epochs and the maximum, mean, minimum
    values encountered:
rotation_reward: ['0.03', '0.31', '0.30', '0.32', '0.34', '0.34', '0.34', '0.34', '0.34', '
    0.32'], Max: 0.36, Mean: 0.32, Min: 0.03
success_rate: ['0.00', '0.83', '1.85', '2.89', '3.72', '4.94', '5.99', '6.90', '7.71', '8.83'
    ], Max: 9.29, Mean: 4.81, Min: 0.00
episode_lengths: ['7.07', '384.30', '378.22', '378.06', '406.11', '415.43', '402.64', '412.50'
    , '384.10', '434.24'], Max: 482.35, Mean: 396.02, Min: 7.07

Please carefully analyze the policy feedback and provide a new, improved reward function that
    can better solve the task. Some helpful tips for analyzing the policy feedback:
    (1) If the success rates are always near zero, then you must rewrite the entire reward
    function
    (2) If the values for a certain reward component are near identical throughout, then this
    means RL is not able to optimize this component as it is written. You may consider
        (a) Changing its scale or the value of its temperature parameter
        (b) Re-writing the reward component
        (c) Discarding the reward component
    (3) If some reward component magnitude is significantly larger, then you must re-scale its
     value to a proper range

Please analyze each existing reward component in the suggested manner above first, and then
    write the reward function code. The output of the reward function should consist of two
    items:
    (1) the total reward,
    (2) a dictionary of each individual reward component.
The code output should be formatted as a python code string: "'''python ... '''".

Some helpful tips for writing the reward function code:
    (1) You may find it helpful to normalize the reward to a fixed range by applying
     transformations like torch.exp to the overall reward or its components
    (2) If you choose to transform a reward component, then you must also introduce a
     temperature parameter inside the transformation function; this parameter must be a named
     variable in the reward function and it must not be an input variable. Each transformed
     reward component should have its own temperature variable
    (3) Make sure the type of each input variable is correctly specified; a float input
     variable should not be specified as torch.Tensor
    (4) Most importantly, the reward code input variables must contain only attributes of the
     provided environment class definition (namely, variables that have prefix self.). Under
     no circumstance can you introduce new input variables.

...

#######################################################################

Iteration 2: Eureka Output:
Based on the analysis of the policy feedback, the rotation_reward component has minimum,
    maximum, and mean values within a fairly close range and seems stable. The success_rate
    has increased gradually, indicating that the agent is learning to some degree. However,
    the episode_lengths have quite some variance, which could indicate that the agent does
    not consistently perform well during training.

Since the rotation_reward component seems mostly adequate, I would recommend introducing
    another component, such as a penalty for large angular velocities, to reinforce stable
    spinning behavior. Here is a new reward function that incorporates an
    angular_velocity_penalty:

'''python
import torch
from typing import Tuple, Dict

@torch.jit.script
def compute_reward(object_rot: torch.Tensor, goal_rot: torch.Tensor, object_angvel: torch.
    Tensor) -> Tuple[torch.Tensor, Dict[str, torch.Tensor]]:
    # Rotation reward
    rot_diff = torch.abs(torch.sum(object_rot * goal_rot, dim=1) - 1) / 2
    rotation_reward = torch.exp(-20 * rot_diff)
    rotation_temp = 20.0

    # Angular velocity penalty
    angvel_norm = torch.norm(object_angvel, dim=1)
    angvel_threshold = 5.0
    angvel_penalty_temp = 10.0
    angular_velocity_penalty = torch.where(angvel_norm > angvel_threshold, torch.exp(-
     angvel_penalty_temp * (angvel_norm - angvel_threshold)), torch.zeros_like(angvel_norm))

    total_reward = rotation_reward - angular_velocity_penalty
```

```
    reward_components = {
        "rotation_reward": rotation_reward,
        "angular_velocity_penalty": angular_velocity_penalty
    }

    return total_reward, reward_components
```

The new reward function applies a penalty for large angular velocities (larger than the
    angvel_threshold) using a temperature parameter (angvel_penalty_temp) to scale the
    penalty. The total reward now combines the rotation_reward with the
    angular_velocity_penalty.

Example 2: EUREKA Reward Reflection on Humanoid. Iteration 2 Score: 3.79, Iteration 3 Score: 4.37

```
Iteration 2: Eureka Output:
Based on the provided feedback, it seems like the `track_reward` component is not contributing
     to the learning process as all its values are 0. Therefore, I will discard the `
     track_reward` and write a new reward function that emphasizes the `velocity_reward`
     component. I will also add a new component to the reward function to encourage the agent
     to maintain upright posture.

Below is the updated reward function:

```python
import torch
from typing import Tuple, Dict

@torch.jit.script
def compute_reward(root_states: torch.Tensor, dt: float) -> Tuple[torch.Tensor, Dict[str,
     torch.Tensor]]:
    # Calculate the forward velocity
    forward_velocity = root_states[:, 7]

    # Reward for forward velocity
    velocity_temperature = 1.0  # You can tune this temperature parameter to adjust how much
     weight the forward velocity gets
    velocity_reward = torch.exp(forward_velocity / velocity_temperature)

    # Calculate the height of the humanoid torso
    torso_height = root_states[:, 2]

    # Reward for maintaining upright posture
    posture_temperature = 1.0  # Tune this temperature parameter to adjust the weight of the
     posture reward component
    posture_reward = torch.exp(torso_height / posture_temperature)

    # Combine rewards
    total_reward = velocity_reward + posture_reward

    reward_components = {'velocity_reward': velocity_reward, 'posture_reward': posture_reward}

    return total_reward, reward_components
```

I have removed the `track_reward` component and added a new component, `posture_reward`, to
     encourage the humanoid to maintain an upright posture. The posture reward is computed as
     the exponential of the humanoid's torso height divided by a temperature parameter. Adjust
      the temperature parameters of both components to fine-tune the model's learning process.

######################################################################

Iteration 2: Reward Reflection:
We trained a RL policy using the provided reward function code and track the values of the
     individual components in the reward function as well as global policy metrics such as
     success rates and episode lengths after every 100 epochs and the maximum, mean, minimum
     values encountered:
velocity_reward: ['1.05', '6.33', '30.09', '66.72', '101.08', '133.44', '148.37', '175.90', '
     188.68', '204.21'], Max: 235.50, Mean: 116.81, Min: 1.05
posture_reward: ['3.26', '3.31', '3.28', '3.27', '3.27', '3.25', '3.28', '3.28', '3.28', '3.27
     '], Max: 3.33, Mean: 3.27, Min: 3.02
consecutive_successes: ['-0.05', '1.18', '2.31', '2.95', '3.20', '3.45', '3.52', '3.61', '3.66
     ', '3.67'], Max: 3.79, Mean: 2.95, Min: -0.05
episode_lengths: ['31.94', '85.50', '104.58', '123.72', '128.38', '130.28', '135.18', '138.20'
     , '134.81', '136.41'], Max: 142.11, Mean: 120.44, Min: 25.89
Please carefully analyze the policy feedback and provide a new, improved reward function that
     can better solve the task. Some helpful tips for analyzing the policy feedback:
    (1) If the success rates are always near zero, then you must rewrite the entire reward
     function
```

```
    (2) If the values for a certain reward component are near identical throughout, then this
    means RL is not able to optimize this component as it is written. You may consider
        (a) Changing its scale or the value of its temperature parameter
        (b) Re-writing the reward component
        (c) Discarding the reward component
    (3) If some reward component magnitude is significantly larger, then you must re-scale its
    value to a proper range
Please analyze each existing reward component in the suggested manner above first, and then
    write the reward function code. The output of the reward function should consist of two
    items:
    (1) the total reward,
    (2) a dictionary of each individual reward component.
The code output should be formatted as a python code string: "'''python ... '''".

Some helpful tips for writing the reward function code:
    (1) You may find it helpful to normalized the reward to a fixed range by applying
    transformations like torch.exp to the overall reward or its components
    (2) If you choose to transform a reward component, then you must also introduce a
    temeperature parameter inside the transformation function; this parameter must be a named
    variable in the reward function and it must not be an input variable. Each transformed
    reward component should have its own temperature variable
    (3) Make sure the type of each input variable is correctly specified; a float input
    variable should not be specified as torch.Tensor
    (4) most importantly, the reward code input variables must contain only attributes of the
    provided environment class definition (namely, variables that have prefix self.). Under
    no circumstance can you introduce new input variables.

#####################################################################

Iteration 3: Eureka Output:
Based on the provided feedback, it seems that the 'posture_reward' component has little impact
    on the learning process, as its values are near identical throughout the training. I
    will modify the reward function to better balance the 'velocity_reward' and '
    posture_reward' components, and also add a new reward component to encourage the agent to
    minimize the energy consumption of its action (i.e., 'action_energy').

Here is the updated reward function:

'''python
import torch
from typing import Tuple, Dict

@torch.jit.script
def compute_reward(root_states: torch.Tensor, actions: torch.Tensor, dt: float) -> Tuple[torch
    .Tensor, Dict[str, torch.Tensor]]:
    # Calculate the forward velocity
    forward_velocity = root_states[:, 7]

    # Reward for forward velocity
    velocity_temperature = 5.0  # I increased the value from 1.0 to 5.0 to balance the impact
     of the velocity reward.
    velocity_reward = torch.exp(forward_velocity / velocity_temperature)

    # Calculate the height of the humanoid torso
    torso_height = root_states[:, 2]

    # Reward for maintaining upright posture
    posture_temperature = 10.0  # Increased the value from 1.0 to 10.0 to balance the impact
     of the posture reward.
    posture_reward = torch.exp(torso_height / posture_temperature)

    # Reward for minimizing energy consumption in actions
    action_energy = torch.norm(actions, p=2, dim=-1)
    energy_temperature = 5.0
    energy_reward = -torch.exp(action_energy / energy_temperature)  # Subtract this component
     to minimize energy consumption.

    # Combine rewards
    total_reward = velocity_reward + posture_reward + energy_reward

    reward_components = {'velocity_reward': velocity_reward, 'posture_reward': posture_reward,
     'energy_reward': energy_reward}

    return total_reward, reward_components
'''

I increased the temperature parameters for both the 'velocity_reward' and 'posture_reward'
    components to balance their impact on the learning process. Additionally, I added a new
    reward component, 'energy_reward', to encourage the agent to minimize the energy
    consumption in its actions. Experiment with different temperature parameter values if
    necessary to fine-tune the policy further.
```

## G.2 NEGATIVELY CORRELATED EUREKA REWARD EXAMPLES

We display several negatively correlated but superior rewards that EUREKA discovers in our experiments.

Example 1: Task: ShadowHand, Human Normalized Score: 1.45, Correlation:−0.26

```python
@torch.jit.script
def compute_reward(object_rot: torch.Tensor, goal_rot: torch.Tensor, fingertip_pos: torch.
    Tensor, object_pos: torch.Tensor) -> Tuple[torch.Tensor, Dict[str, torch.Tensor]]:
    # Compute the quaternion distance between the object's current orientation and the goal
     orientation
    q_dist = torch.sum((object_rot * goal_rot), dim=-1)
    q_dist = torch.min(q_dist, 1 - q_dist)  # Make sure the q_dist is in the range [0, 1]

    # Normalize the quaternion distance using a temperature parameter
    temp_rot = 0.5
    rot_reward = torch.exp(-temp_rot * q_dist)

    # Compute the distance between the fingertips and the object center
    fingertips_object_dist = torch.norm(fingertip_pos - object_pos[:, None], dim=-1)

    # Apply a threshold for the distance
    distance_threshold = 0.1
    close_enough = (fingertips_object_dist < distance_threshold).to(torch.float32)

    # Normalize the distance between fingertips and object center using an updated temperature
      parameter
    temp_dist = 10.0
    distance_reward = torch.mean(torch.exp(-temp_dist * fingertips_object_dist * close_enough)
     , dim=-1)

    # Apply a penalty if the agent is not close enough to the object
    distance_penalty = 0.5 * (1 - torch.prod(close_enough, dim=-1))

    # Combine the reward components
    total_reward = rot_reward * distance_reward - distance_penalty

    # Store the reward components in a dictionary
    reward_components = {
        "rot_reward": rot_reward,
        "distance_reward": distance_reward,
        "distance_penalty": distance_penalty,
        "total_reward": total_reward,
    }

    return total_reward, reward_components
```

Example 2: Task: FrankaCabinet, Human Normalized Score: 11.98, Correlation:−0.30

```python
@torch.jit.script
def compute_reward(franka_grasp_pos: torch.Tensor, drawer_grasp_pos: torch.Tensor,
    cabinet_dof_pos: torch.Tensor,
                  franka_lfinger_pos: torch.Tensor, franka_rfinger_pos: torch.Tensor) ->
    Tuple[torch.Tensor, Dict[str, torch.Tensor]]:

    # Calculate the distance between the Franka grasping position and the cabinet grasping
     position
    grasp_distance = torch.norm(franka_grasp_pos - drawer_grasp_pos, dim=-1)

    # Calculate the distances between franke_lfinger_pos, franka_rfinger_pos and
     drawer_grasp_pos
    lfinger_distance = torch.norm(franka_lfinger_pos - drawer_grasp_pos, dim=-1)
    rfinger_distance = torch.norm(franka_rfinger_pos - drawer_grasp_pos, dim=-1)

    # Calculate the drawer opening distance
    drawer_opening = cabinet_dof_pos[:, 3]

    # Define temperature parameters for transforming the reward components
    grasp_distance_scaling = torch.tensor(20.0)
    handle_grasping_temperature = torch.tensor(20.0)
    drawer_opening_temperature = torch.tensor(20.0)

    # Transform the reward components
    grasp_distance_reward = 1.0 / (1.0 + grasp_distance_scaling * grasp_distance)
```

```
        handle_grasping_reward = torch.exp(-handle_grasping_temperature * (lfinger_distance +
        rfinger_distance))
        drawer_opening_reward = torch.exp(drawer_opening_temperature * drawer_opening)

        # Compute the total reward
        reward = grasp_distance_reward + handle_grasping_reward + drawer_opening_reward

        # Create a dictionary of individual reward components
        reward_components = {
            "grasp_distance_reward": grasp_distance_reward,
            "handle_grasping_reward": handle_grasping_reward,
            "drawer_opening_reward": drawer_opening_reward
        }

        return reward, reward_components
```

## G.3 EUREKA FROM HUMAN INITIALIZATION EXAMPLES

We display several examples of a single step in the EUREKA from Human Initialization setting. In these examples, the first reward (Iteration 0) is the original human-written task reward, and Iteration 1 is the best reward after one step of EUREKA improvement.

Example 1: EUREKA Human Initialization on Kettle. Human Success Rate: 0.11, EUREKA Success Rate: 0.91

```
Iteration 1: Human Initialization:
```python
import torch

@torch.jit.script
def compute_reward(
    kettle_handle_pos, bucket_handle_pos, kettle_spout_pos,
    right_hand_ff_pos, right_hand_mf_pos, right_hand_rf_pos, right_hand_lf_pos,
     right_hand_th_pos,
    left_hand_ff_pos, left_hand_mf_pos, left_hand_rf_pos, left_hand_lf_pos, left_hand_th_pos,
):
    right_hand_finger_dist = (torch.norm(kettle_handle_pos - right_hand_ff_pos, p=2, dim=-1) +
     torch.norm(kettle_handle_pos - right_hand_mf_pos, p=2, dim=-1)
                            + torch.norm(kettle_handle_pos - right_hand_rf_pos, p=2, dim=-1) +
     torch.norm(kettle_handle_pos - right_hand_lf_pos, p=2, dim=-1)
                            + torch.norm(kettle_handle_pos - right_hand_th_pos, p=2, dim=-1))
    left_hand_finger_dist = (torch.norm(bucket_handle_pos - left_hand_ff_pos, p=2, dim=-1) +
     torch.norm(bucket_handle_pos - left_hand_mf_pos, p=2, dim=-1)
                            + torch.norm(bucket_handle_pos - left_hand_rf_pos, p=2, dim=-1) +
     torch.norm(bucket_handle_pos - left_hand_lf_pos, p=2, dim=-1)
                            + torch.norm(bucket_handle_pos - left_hand_th_pos, p=2, dim=-1))

    right_hand_dist_rew = right_hand_finger_dist
    left_hand_dist_rew = left_hand_finger_dist

    up_rew = torch.zeros_like(right_hand_dist_rew)
    up_rew = torch.where(right_hand_finger_dist < 0.7,
                    torch.where(left_hand_finger_dist < 0.7,
                                    0.5 - torch.norm(bucket_handle_pos - kettle_spout_pos, p
     =2, dim=-1) * 2, up_rew), up_rew)

    reward = 1 + up_rew - right_hand_dist_rew - left_hand_dist_rew
    reward_components = {
        'up_reward': up_rew,
        'right_hand_dist_reward': right_hand_dist_rew,
        'left_hand_dist_reward': left_hand_dist_rew,
    }

    return reward, reward_components
```

########################################################################

Iteration 1: Reward Reflection:
We trained a RL policy using the provided reward function code and tracked the values of the
     individual components in the reward function as well as global policy metrics such as
     success rates and episode lengths after every 300 epochs and the maximum, mean, minimum
     values encountered:
up_reward: ['0.00', '0.00', '0.02', '-0.01', '0.04', '-0.02', '0.03', '-0.01', '0.02', '0.04'
     ], Max: 0.12, Mean: 0.03, Min: -0.06
right_hand_dist_reward: ['0.93', '0.34', '0.40', '0.38', '0.36', '0.44', '0.43', '0.42', '0.38
     ', '0.42'], Max: 0.96, Mean: 0.41, Min: 0.27
left_hand_dist_reward: ['1.47', '0.87', '0.69', '0.89', '0.83', '0.81', '0.83', '1.02', '0.84'
     , '0.85'], Max: 3.57, Mean: 0.88, Min: 0.39
```

```
consecutive_successes: ['0.00', '0.00', '0.00', '0.00', '0.00', '0.00', '0.00', '0.00', '0.00'
    , '0.00'], Max: 0.01, Mean: 0.00, Min: 0.00
episode_lengths: ['125.00', '122.45', '123.92', '124.24', '124.22', '124.83', '124.86', '
    124.42', '124.89', '124.18'], Max: 125.00, Mean: 123.81, Min: 111.56
Please carefully analyze the policy feedback and provide a new, improved reward function that
    can better solve the task. Some helpful tips for analyzing the policy feedback:
    (1) If the success rates are always near zero, then you must rewrite the entire reward
    function
    (2) If the values for a certain reward component are near identical throughout, then this
    means RL is not able to optimize this component as it is written. You may consider
        (a) Changing its scale or the value of its temperature parameter
        (b) Re-writing the reward component
        (c) Discarding the reward component
    (3) If some reward component magnitude is significantly larger, then you must re-scale its
     value to a proper range
Please analyze each existing reward component in the suggested manner above first, and then
    write the reward function code. The output of the reward function should consist of two
    items:
    (1) the total reward,
    (2) a dictionary of each individual reward component.
The code output should be formatted as a python code string: "```python ... ```".

Some helpful tips for writing the reward function code:
    (1) You may find it helpful to normalize the reward to a fixed range by applying
    transformations like torch.exp to the overall reward or its components
    (2) If you choose to transform a reward component, then you must also introduce a
    temperature parameter inside the transformation function; this parameter must be a named
    variable in the reward function and it must not be an input variable. Each transformed
    reward component should have its own temperature variable
    (3) Make sure the type of each input variable is correctly specified; a float input
    variable should not be specified as torch.Tensor
    (4) Most importantly, the reward code input variables must contain only attributes of the
    provided environment class definition (namely, variables that have prefix self.). Under
    no circumstance can you introduce new input variables.

########################################################################

Iteration 2: Eureka Output:
From the feedback data you provided, it is evident that up_reward has a rather small value and
     success rate is always near zero. We can try some modifications to improve the reward
    function. First, we can update the distance rewards by adjusting their scale and applying
     an exponential transformation with an associated temperature. This will make their
    values range between 0 and 1. We will also slightly adjust the up_reward to get more
    value when the kettle and bucket are closer. Finally, we will add some weights to the
    reward components to balance their contribution to the total reward.

Here is the updated reward function:

```python
import torch

@torch.jit.script
def compute_reward(
    kettle_handle_pos, bucket_handle_pos, kettle_spout_pos,
    right_hand_ff_pos, right_hand_mf_pos, right_hand_rf_pos, right_hand_lf_pos,
     right_hand_th_pos,
    left_hand_ff_pos, left_hand_mf_pos, left_hand_rf_pos, left_hand_lf_pos, left_hand_th_pos,
):
    dist_temp = torch.tensor(0.1)

    right_hand_finger_dist = (torch.norm(kettle_handle_pos - right_hand_ff_pos, p=2, dim=-1) +
      torch.norm(kettle_handle_pos - right_hand_mf_pos, p=2, dim=-1)
                            + torch.norm(kettle_handle_pos - right_hand_rf_pos, p=2, dim=-1) +
      torch.norm(kettle_handle_pos - right_hand_lf_pos, p=2, dim=-1)
                            + torch.norm(kettle_handle_pos - right_hand_th_pos, p=2, dim=-1))
    left_hand_finger_dist = (torch.norm(bucket_handle_pos - left_hand_ff_pos, p=2, dim=-1) +
     torch.norm(bucket_handle_pos - left_hand_mf_pos, p=2, dim=-1)
                            + torch.norm(bucket_handle_pos - left_hand_rf_pos, p=2, dim=-1) +
     torch.norm(bucket_handle_pos - left_hand_lf_pos, p=2, dim=-1)
                            + torch.norm(bucket_handle_pos - left_hand_th_pos, p=2, dim=-1))

    right_hand_dist_rew = 1 - torch.exp(-dist_temp * right_hand_finger_dist)
    left_hand_dist_rew = 1 - torch.exp(-dist_temp * left_hand_finger_dist)

    up_rew = torch.zeros_like(right_hand_dist_rew)
    up_rew = torch.where(right_hand_finger_dist < 0.7,
                    torch.where(left_hand_finger_dist < 0.7,
                                    1 - torch.norm(bucket_handle_pos - kettle_spout_pos, p=2,
     dim=-1) * 2, up_rew), up_rew)

    reward = 5 * up_rew - 3 * right_hand_dist_rew - 2 * left_hand_dist_rew
```
```

```
    reward_components = {
        'up_reward': up_rew,
        'right_hand_dist_reward': right_hand_dist_rew,
        'left_hand_dist_reward': left_hand_dist_rew,
    }

    return reward, reward_components
```

Please test this updated reward function and provide new policy feedback data for further
    analysis if needed.
```

Example 2: EUREKA Human Initialization on CatchUnderarm. Human Success Rate: 0.33, EUREKA Success
Rate: 0.64

```
Iteration 1: Human Initialization:
```python
import torch
from isaacgym.torch_utils import quat_mul, quat_conjugate

@torch.jit.script
def compute_reward(
    object_pos, object_rot, goal_pos, goal_rot,
):

    goal_dist = torch.norm(goal_pos - object_pos, p=2, dim=-1)

    quat_diff = quat_mul(object_rot, quat_conjugate(goal_rot))
    rot_dist = 2.0 * torch.asin(torch.clamp(torch.norm(quat_diff[:, 0:3], p=2, dim=-1), max
     =1.0))

    dist_rew = goal_dist
    reward = torch.exp(-0.2*(dist_rew * 50 + rot_dist))
    reward_components = {
        "dist_reward": dist_rew,
        "rot_reward": rot_dist,
    }

    return reward, reward_components
```

#######################################################################

Iteration 1: Reward Reflection:
We trained a RL policy using the provided reward function code and tracked the values of the
    individual components in the reward function as well as global policy metrics such as
    success rates and episode lengths after every 300 epochs and the maximum, mean, minimum
    values encountered:
dist_reward: ['0.41', '0.34', '0.34', '0.29', '0.26', '0.25', '0.23', '0.21', '0.21', '0.21'],
     Max: 0.42, Mean: 0.26, Min: 0.19
rot_reward: ['2.21', '2.21', '2.19', '2.22', '2.20', '2.21', '2.22', '2.21', '2.20', '2.20'],
    Max: 2.27, Mean: 2.21, Min: 2.16
consecutive_successes: ['0.00', '0.00', '0.00', '0.00', '0.04', '0.14', '0.25', '0.17', '0.14'
    , '0.03'], Max: 0.33, Mean: 0.08, Min: 0.00
episode_lengths: ['28.00', '59.83', '56.22', '51.10', '56.34', '57.48', '61.40', '64.13', '
    66.47', '67.11'], Max: 75.00, Mean: 60.98, Min: 28.00
Please carefully analyze the policy feedback and provide a new, improved reward function that
    can better solve the task. Some helpful tips for analyzing the policy feedback:
    (1) If the success rates are always near zero, then you must rewrite the entire reward
    function
    (2) If the values for a certain reward component are near identical throughout, then this
    means RL is not able to optimize this component as it is written. You may consider
        (a) Changing its scale or the value of its temperature parameter
        (b) Re-writing the reward component
        (c) Discarding the reward component
    (3) If some reward component magnitude is significantly larger, then you must re-scale its
     value to a proper range
Please analyze each existing reward component in the suggested manner above first, and then
    write the reward function code. The output of the reward function should consist of two
    items:
    (1) the total reward,
    (2) a dictionary of each individual reward component.
The code output should be formatted as a python code string: "```python ... ```".

Some helpful tips for writing the reward function code:
    (1) You may find it helpful to normalize the reward to a fixed range by applying
    transformations like torch.exp to the overall reward or its components
    (2) If you choose to transform a reward component, then you must also introduce a
    temperature parameter inside the transformation function; this parameter must be a named
```

```
        variable in the reward function and it must not be an input variable. Each transformed
        reward component should have its own temperature variable
        (3) Make sure the type of each input variable is correctly specified; a float input
        variable should not be specified as torch.Tensor
        (4) Most importantly, the reward code input variables must contain only attributes of the
        provided environment class definition (namely, variables that have prefix self.). Under
        no circumstance can you introduce new input variables.

########################################################################

Iteration 2: Eureka Output:
Based on the policy feedback, we can observe that the success rates and episode lengths are
     improving over time. However, the values for `rot_reward` are near identical throughout
     policy learning, which means the RL agent is not able to optimize the current rotation
     reward component. We can change the scale/temperature parameter of the rotation reward
     and try to make it more sensitive to the changes in rotation. We will keep the distance
     reward component as it is because it seems to be optimized during training.

Here is an updated reward function for the given task:

```python
import torch
from isaacgym.torch_utils import quat_mul, quat_conjugate

@torch.jit.script
def compute_reward(
    object_pos: torch.Tensor, object_rot: torch.Tensor,
    goal_pos: torch.Tensor, goal_rot: torch.Tensor
) -> Tuple[torch.Tensor, Dict[str, torch.Tensor]]:

    goal_dist = torch.norm(goal_pos - object_pos, p=2, dim=-1)

    quat_diff = quat_mul(object_rot, quat_conjugate(goal_rot))
    rot_dist = 2.0 * torch.asin(torch.clamp(torch.norm(quat_diff[:, 0:3], p=2, dim=-1), max
     =1.0))

    dist_rew = goal_dist
    rot_rew_temp = 0.1
    rot_rew = torch.exp(-rot_rew_temp * rot_dist)

    reward = torch.exp(-0.25 * dist_rew * 50) * rot_rew

    reward_components = {
        "dist_reward": dist_rew,
        "rot_reward": rot_dist,
    }

    return reward, reward_components
```
The new reward function changes the scale of the rotation reward component by introducing a
     temperature parameter `rot_rew_temp` and applying a transformation using `torch.exp`.
     This should help the RL agent better optimize the rotation reward component during policy
      learning.
```

## G.4 EUREKA FROM HUMAN REWARD REFLECTION

We display the raw dialogue that includes all human reward reflection texts as well as the generated
EUREKA rewards in our EUREKA from human reward reflection experiment.

Example 1: EUREKA from Human Reward Reflection

```
########################################################################

Iteration 1: Eureka Output:

The task is to make the humanoid run as fast as possible.
Therefore, the reward function should encourage forward movement.
This can be achieved by rewarding the agent based on the velocity in the forward direction.
We can also penalize the agent for using too much energy, as this would encourage efficient
     movement.

Here is a possible reward function:

```python
def compute_humanoid_reward(obs_buf, reset_buf, progress_buf, actions, up_weight,
     heading_weight, potentials, prev_potentials, actions_cost_scale, energy_cost_scale,
     joints_at_limit_cost_scale, max_motor_effort, motor_efforts, termination_height,
     death_cost, max_episode_length):
```

```
    # type: (Tensor, Tensor, Tensor, Tensor, float, float, Tensor, Tensor, float, float, float
    , float, Tensor, float, float, float) -> Tuple[Tensor, Tensor]

    # Reward is based on forward velocity
    reward = obs_buf[:, 1]

    # Penalize energy usage
    energy_cost = energy_cost_scale * torch.sum(actions**2, dim=-1)
    reward -= energy_cost

    # Penalize joints at limit
    joints_at_limit_cost = joints_at_limit_cost_scale * torch.sum(obs_buf[:, 12:32] > 0.99,
     dim=-1)
    reward -= joints_at_limit_cost

    # Penalize for falling
    not_upright = torch.abs(1 - obs_buf[:, 10]) > 0.25
    reward = torch.where(not_upright, torch.ones_like(reward) * death_cost, reward)

    # Reset if fallen
    reset = torch.where(not_upright, torch.ones_like(reset_buf), reset_buf)
    reset = torch.where(progress_buf >= max_episode_length - 1, torch.ones_like(reset_buf),
     reset)

    return reward, reset
```

This reward function encourages the agent to move forward as fast as possible, while also
    encouraging efficient movement and penalizing the agent for falling or reaching joint
    limits.

########################################################################

Iteration 1: Human Reward Reflection:
The learned behavior resembles forward squat jump;
please revise the reward function so that the behavior resembles forward running.

########################################################################

Iteration 2: Eureka Output:
 To encourage a running behavior, we can modify the reward function to reward the agent for
    alternating leg movements and penalize for jumping (both feet off the ground).
  We can also reward the agent for maintaining a relatively upright torso to resemble a
    running posture.

Here is a revised reward function:

```python
def compute_humanoid_reward(obs_buf, reset_buf, progress_buf, actions, up_weight,
    heading_weight, potentials, prev_potentials, actions_cost_scale, energy_cost_scale,
    joints_at_limit_cost_scale, max_motor_effort, motor_efforts, termination_height,
    death_cost, max_episode_length):
    # type: (Tensor, Tensor, Tensor, Tensor, float, float, Tensor, Tensor, float, float, float
    , float, Tensor, float, float, float) -> Tuple[Tensor, Tensor]

    # Reward is based on forward velocity
    reward = obs_buf[:, 1]

    # Penalize energy usage
    energy_cost = energy_cost_scale * torch.sum(actions**2, dim=-1)
    reward -= energy_cost

    # Penalize joints at limit
    joints_at_limit_cost = joints_at_limit_cost_scale * torch.sum(obs_buf[:, 12:32] > 0.99,
     dim=-1)
    reward -= joints_at_limit_cost

    # Reward for alternating leg movements
    leg_difference = torch.abs(obs_buf[:, 12] - obs_buf[:, 13])
    reward += leg_difference

    # Penalize for jumping (both feet off the ground)
    feet_off_ground = torch.sum(obs_buf[:, 54:56] == 0)
    reward -= feet_off_ground

    # Reward for maintaining an upright torso
    upright_torso = 1 - torch.abs(1 - obs_buf[:, 10])
    reward += upright_torso

    # Penalize for falling
    not_upright = torch.abs(1 - obs_buf[:, 10]) > 0.25
```

```
    reward = torch.where(not_upright, torch.ones_like(reward) * death_cost, reward)

    # Reset if fallen
    reset = torch.where(not_upright, torch.ones_like(reset_buf), reset_buf)
    reset = torch.where(progress_buf >= max_episode_length - 1, torch.ones_like(reset_buf),
     reset)

    return reward, reset
```

This reward function encourages the agent to move forward as fast as possible, while also
    encouraging efficient movement, alternating leg movements, maintaining an upright torso,
    and penalizing the agent for jumping or reaching joint limits.

#######################################################################

Iteration 2: Human Reward Reflection:
The learned behavior now looks like duck walk;
the legs are indeed alternating but the torso is very low.
Could you improve the reward function for upright running?

#######################################################################

Iteration 3: Eureka Output:
To encourage a more upright running posture,
  we can increase the reward for maintaining an upright torso and penalize the agent more
    heavily for having a low torso position.
  We can also add a reward for maintaining a certain height,
  which would discourage the "duck walk" behavior.

Here is a revised reward function:

```python
def compute_humanoid_reward(obs_buf, reset_buf, progress_buf, actions, up_weight,
     heading_weight, potentials, prev_potentials, actions_cost_scale, energy_cost_scale,
     joints_at_limit_cost_scale, max_motor_effort, motor_efforts, termination_height,
     death_cost, max_episode_length):
    # type: (Tensor, Tensor, Tensor, Tensor, float, float, Tensor, Tensor, float, float, float
    , float, Tensor, float, float, float) -> Tuple[Tensor, Tensor]

    # Reward is based on forward velocity
    reward = obs_buf[:, 1]

    # Penalize energy usage
    energy_cost = energy_cost_scale * torch.sum(actions**2, dim=-1)
    reward -= energy_cost

    # Penalize joints at limit
    joints_at_limit_cost = joints_at_limit_cost_scale * torch.sum(obs_buf[:, 12:32] > 0.99,
     dim=-1)
    reward -= joints_at_limit_cost

    # Reward for alternating leg movements
    leg_difference = torch.abs(obs_buf[:, 12] - obs_buf[:, 13])
    reward += leg_difference

    # Penalize for jumping (both feet off the ground)
    feet_off_ground = torch.sum(obs_buf[:, 54:56] == 0)
    reward -= feet_off_ground

    # Reward for maintaining an upright torso
    upright_torso = 1 - torch.abs(1 - obs_buf[:, 10])
    reward += 2 * upright_torso  # Increase reward for upright torso

    # Penalize for low torso position
    low_torso = obs_buf[:, 0] < 1.0
    reward = torch.where(low_torso, torch.ones_like(reward) * -1.0, reward)

    # Reward for maintaining a certain height
    height_reward = obs_buf[:, 0]
    reward += height_reward

    # Penalize for falling
    not_upright = torch.abs(1 - obs_buf[:, 10]) > 0.25
    reward = torch.where(not_upright, torch.ones_like(reward) * death_cost, reward)

    # Reset if fallen
    reset = torch.where(not_upright, torch.ones_like(reset_buf), reset_buf)
    reset = torch.where(progress_buf >= max_episode_length - 1, torch.ones_like(reset_buf),
     reset)
```

```
    return reward, reset
```

This reward function encourages the agent to move forward as fast as possible, while also
    encouraging efficient movement, alternating leg movements, maintaining an upright torso,
    maintaining a certain height, and penalizing the agent for jumping, having a low torso
    position, or reaching joint limits.

#######################################################################

Iteration 3: Human Reward Reflection:
The learned behavior has the robot hopping on one of its foot in order to move forward.
 Please revise the reward function to encourage upright running behavior.

#######################################################################

Iteration 4: Eureka Output:
 To discourage hopping and encourage a more balanced running behavior,
  we can add a penalty for not using both legs equally.
   We can also increase the reward for maintaining an upright torso and add a penalty for high
      angular velocity,
    which would discourage spinning or hopping.

Here is a revised reward function:

```python
def compute_humanoid_reward(obs_buf, reset_buf, progress_buf, actions, up_weight,
    heading_weight, potentials, prev_potentials, actions_cost_scale, energy_cost_scale,
    joints_at_limit_cost_scale, max_motor_effort, motor_efforts, termination_height,
    death_cost, max_episode_length):
    # type: (Tensor, Tensor, Tensor, Tensor, float, float, Tensor, Tensor, float, float, float
    , float, Tensor, float, float, float) -> Tuple[Tensor, Tensor]

    # Reward is based on forward velocity
    reward = obs_buf[:, 1]

    # Penalize energy usage
    energy_cost = energy_cost_scale * torch.sum(actions**2, dim=-1)
    reward -= energy_cost

    # Penalize joints at limit
    joints_at_limit_cost = joints_at_limit_cost_scale * torch.sum(obs_buf[:, 12:32] > 0.99,
     dim=-1)
    reward -= joints_at_limit_cost

    # Reward for alternating leg movements
    leg_difference = torch.abs(obs_buf[:, 12] - obs_buf[:, 13])
    reward += leg_difference

    # Penalize for not using both legs equally
    leg_usage_difference = torch.abs(torch.sum(obs_buf[:, 12:14], dim=-1) - 1.0)
    reward -= leg_usage_difference

    # Penalize for jumping (both feet off the ground)
    feet_off_ground = torch.sum(obs_buf[:, 54:56] == 0)
    reward -= feet_off_ground

    # Reward for maintaining an upright torso
    upright_torso = 1 - torch.abs(1 - obs_buf[:, 10])
    reward += 2 * upright_torso  # Increase reward for upright torso

    # Penalize for high angular velocity
    high_ang_velocity = torch.abs(obs_buf[:, 4:7]).sum(dim=-1) > 1.0
    reward = torch.where(high_ang_velocity, torch.ones_like(reward) * -1.0, reward)

    # Penalize for falling
    not_upright = torch.abs(1 - obs_buf[:, 10]) > 0.25
    reward = torch.where(not_upright, torch.ones_like(reward) * death_cost, reward)

    # Reset if fallen
    reset = torch.where(not_upright, torch.ones_like(reset_buf), reset_buf)
    reset = torch.where(progress_buf >= max_episode_length - 1, torch.ones_like(reset_buf),
     reset)

    return reward, reset
```

This reward function encourages the agent to move forward as fast as possible, while also
    encouraging efficient movement, alternating leg movements, maintaining an upright torso,
    and penalizing the agent for jumping, not using both legs equally, having high angular
    velocity, or reaching joint limits.

```
########################################################################

Iteration 4: Human Reward Reflection:
This reward function removed the penalty for low torse position that you added last time;
    could you just add it back in?

########################################################################

Iteration 5: Eureka Output:
 Sure, I apologize for the oversight. We can add back the penalty for low torso position to
    discourage the "duck walk" behavior. Here is the revised reward function:

```python
def compute_humanoid_reward(obs_buf, reset_buf, progress_buf, actions, up_weight,
    heading_weight, potentials, prev_potentials, actions_cost_scale, energy_cost_scale,
    joints_at_limit_cost_scale, max_motor_effort, motor_efforts, termination_height,
    death_cost, max_episode_length):
    # type: (Tensor, Tensor, Tensor, Tensor, float, float, Tensor, Tensor, float, float, float
    , float, Tensor, float, float, float) -> Tuple[Tensor, Tensor]

    # Reward is based on forward velocity
    reward = obs_buf[:, 1]

    # Penalize energy usage
    energy_cost = energy_cost_scale * torch.sum(actions**2, dim=-1)
    reward -= energy_cost

    # Penalize joints at limit
    joints_at_limit_cost = joints_at_limit_cost_scale * torch.sum(obs_buf[:, 12:32] > 0.99,
     dim=-1)
    reward -= joints_at_limit_cost

    # Reward for alternating leg movements
    leg_difference = torch.abs(obs_buf[:, 12] - obs_buf[:, 13])
    reward += leg_difference

    # Penalize for not using both legs equally
    leg_usage_difference = torch.abs(torch.sum(obs_buf[:, 12:14], dim=-1) - 1.0)
    reward -= leg_usage_difference

    # Penalize for jumping (both feet off the ground)
    feet_off_ground = torch.sum(obs_buf[:, 54:56] == 0)
    reward -= feet_off_ground

    # Reward for maintaining an upright torso
    upright_torso = 1 - torch.abs(1 - obs_buf[:, 10])
    reward += 2 * upright_torso  # Increase reward for upright torso

    # Penalize for low torso position
    low_torso = obs_buf[:, 0] < 1.0
    reward = torch.where(low_torso, torch.ones_like(reward) * -1.0, reward)

    # Penalize for high angular velocity
    high_ang_velocity = torch.abs(obs_buf[:, 4:7]).sum(dim=-1) > 1.0
    reward = torch.where(high_ang_velocity, torch.ones_like(reward) * -1.0, reward)

    # Penalize for falling
    not_upright = torch.abs(1 - obs_buf[:, 10]) > 0.25
    reward = torch.where(not_upright, torch.ones_like(reward) * death_cost, reward)

    # Reset if fallen
    reset = torch.where(not_upright, torch.ones_like(reset_buf), reset_buf)
    reset = torch.where(progress_buf >= max_episode_length - 1, torch.ones_like(reset_buf),
     reset)

    return reward, reset
```

This reward function encourages the agent to move forward as fast as possible, while also
    encouraging efficient movement, alternating leg movements, maintaining an upright torso,
    and penalizing the agent for jumping, not using both legs equally, having a low torso
    position, having high angular velocity, or reaching joint limits.
```

## G.5 EUREKA AND HUMAN REWARD COMPARISON

We display the human reward on a Dexterity task verbatim and contrast it with a EUREKA-generated
reward on the same task. As shown, the human reward is difficult to parse and has many commented-

out blocks of reward components, suggesting history of trial-and-error reward design. In contrast, EUREKA reward is clean and interpretable, amenable to post-hoc human inspection and editing.

Example 1: Human reward for PushBlock

```python
@torch.jit.script
def compute_hand_reward(
    rew_buf, reset_buf, reset_goal_buf, progress_buf, successes, consecutive_successes,
    max_episode_length: float, object_pos, object_rot, left_target_pos, left_target_rot,
     right_target_pos, right_target_rot, block_right_handle_pos, block_left_handle_pos,
    left_hand_pos, right_hand_pos, right_hand_ff_pos, right_hand_mf_pos, right_hand_rf_pos,
     right_hand_lf_pos, right_hand_th_pos,
    left_hand_ff_pos, left_hand_mf_pos, left_hand_rf_pos, left_hand_lf_pos, left_hand_th_pos,
    dist_reward_scale: float, rot_reward_scale: float, rot_eps: float,
    actions, action_penalty_scale: float,
    success_tolerance: float, reach_goal_bonus: float, fall_dist: float,
    fall_penalty: float, max_consecutive_successes: int, av_factor: float, ignore_z_rot: bool
):
    # Distance from the hand to the object
    left_goal_dist = torch.norm(left_target_pos - block_left_handle_pos, p=2, dim=-1)
    right_goal_dist = torch.norm(right_target_pos - block_right_handle_pos, p=2, dim=-1)
    # goal_dist = target_pos[:, 2] - object_pos[:, 2]

    right_hand_dist = torch.norm(block_right_handle_pos - right_hand_pos, p=2, dim=-1)
    left_hand_dist = torch.norm(block_left_handle_pos - left_hand_pos, p=2, dim=-1)

    right_hand_finger_dist = (torch.norm(block_right_handle_pos - right_hand_ff_pos, p=2, dim
     =-1) + torch.norm(block_right_handle_pos - right_hand_mf_pos, p=2, dim=-1)
                            + torch.norm(block_right_handle_pos - right_hand_rf_pos, p=2, dim
     =-1) + torch.norm(block_right_handle_pos - right_hand_lf_pos, p=2, dim=-1)
                            + torch.norm(block_right_handle_pos - right_hand_th_pos, p=2, dim
     =-1))
    left_hand_finger_dist = (torch.norm(block_left_handle_pos - left_hand_ff_pos, p=2, dim=-1)
      + torch.norm(block_left_handle_pos - left_hand_mf_pos, p=2, dim=-1)
                            + torch.norm(block_left_handle_pos - left_hand_rf_pos, p=2, dim
     =-1) + torch.norm(block_left_handle_pos - left_hand_lf_pos, p=2, dim=-1)
                            + torch.norm(block_left_handle_pos - left_hand_th_pos, p=2, dim
     =-1))
    # Orientation alignment for the cube in hand and goal cube
    # quat_diff = quat_mul(object_rot, quat_conjugate(target_rot))
    # rot_dist = 2.0 * torch.asin(torch.clamp(torch.norm(quat_diff[:, 0:3], p=2, dim=-1), max
     =1.0))

    right_hand_dist_rew = 1.2-1*right_hand_finger_dist
    left_hand_dist_rew = 1.2-1*left_hand_finger_dist

    # rot_rew = 1.0/(torch.abs(rot_dist) + rot_eps) * rot_reward_scale

    action_penalty = torch.sum(actions ** 2, dim=-1)

    # Total reward is: position distance + orientation alignment + action regularization +
     success bonus + fall penalty
    # reward = torch.exp(-0.05*(up_rew * dist_reward_scale)) + torch.exp(-0.05*(
     right_hand_dist_rew * dist_reward_scale)) + torch.exp(-0.05*(left_hand_dist_rew *
     dist_reward_scale))
    up_rew = torch.zeros_like(right_hand_dist_rew)
    up_rew = 5 - 5*left_goal_dist - 5*right_goal_dist

    # reward = torch.exp(-0.1*(right_hand_dist_rew * dist_reward_scale)) + torch.exp(-0.1*(
     left_hand_dist_rew * dist_reward_scale))
    reward = right_hand_dist_rew + left_hand_dist_rew + up_rew

    resets = torch.where(right_hand_finger_dist >= 1.2, torch.ones_like(reset_buf), reset_buf)
    resets = torch.where(left_hand_finger_dist >= 1.2, torch.ones_like(resets), resets)

    # Find out which envs hit the goal and update successes count
    successes = torch.where(successes == 0,
                    torch.where(torch.abs(left_goal_dist) <= 0.1,
                        torch.where(torch.abs(right_goal_dist) <= 0.1, torch.ones_like(
     successes), torch.ones_like(successes) * 0.5), successes), successes)

    resets = torch.where(progress_buf >= max_episode_length, torch.ones_like(resets), resets)

    goal_resets = torch.zeros_like(resets)

    num_resets = torch.sum(resets)
    finished_cons_successes = torch.sum(successes * resets.float())

    cons_successes = torch.where(resets > 0, successes * resets, consecutive_successes).mean()
    # reward = successes
```

```
    return reward, resets, goal_resets, progress_buf, successes, cons_successes
```

Example 2: Eureka reward for PushBlock

```python
@torch.jit.script
def compute_reward(object_pos: Tensor,
                   left_hand_pos: Tensor,
                   right_hand_pos: Tensor,
                   left_goal_pos: Tensor,
                   right_goal_pos: Tensor) -> Tuple[Tensor, Dict[str, Tensor]]:

    # Temperature parameters for reward components
    temp_left_proximity: float = 0.5
    temp_right_proximity: float = 0.5
    temp_hand_distance: float = 0.1

    # Proximity reward for pushing the block to the left goal
    left_push_distance = torch.norm(object_pos - left_goal_pos, dim=1)
    left_proximity_reward = torch.exp(-temp_left_proximity * left_push_distance)

    # Proximity reward for pushing the block to the right goal
    right_push_distance = torch.norm(object_pos - right_goal_pos, dim=1)
    right_proximity_reward = torch.exp(-temp_right_proximity * right_push_distance)

    # Proximity rewards for the hands to be close to the block for better control
    left_hand_to_block = torch.norm(object_pos - left_hand_pos, dim=1)
    left_hand_reward = torch.exp(-temp_hand_distance * left_hand_to_block)

    right_hand_to_block = torch.norm(object_pos - right_hand_pos, dim=1)
    right_hand_reward = torch.exp(-temp_hand_distance * right_hand_to_block)

    # Final reward as a weighted sum of individual reward components
    reward = 0.25 * (left_proximity_reward + right_proximity_reward) + 0.25 * (
     left_hand_reward + right_hand_reward)

    reward_components = {
        "left_proximity_reward": left_proximity_reward,
        "right_proximity_reward": right_proximity_reward,
        "left_hand_reward": left_hand_reward,
        "right_hand_reward": right_hand_reward,
    }

    return reward, reward_components
```

