# OpenReview forum: "Eureka: Human-Level Reward Design via Coding Large Language Models"
_robot-learning.org/CoRL/2023/Workshop/TGR — CoRL 2023 Workshop TGR Oral_

### Official Review · Reviewer_9WQy · 2023-10-16

**Rating:** 7
**Confidence:** 3

**Review:**

Reward designing could be time-consuming and ad-hoc for different tasks. This paper automate this process with LLM. It provides a promising way to make robots adapts to different applications. This paper and its approach fit this workshop's topic very well.

---

### Decision · Program_Chairs · 2023-10-20

**Decision:**

Accept (Oral)

**Comment:**

Very cool results, and highly relevant to the workshop topic!